# Photodynamic inactivation of multidrug-resistant strains of *Klebsiella pneumoniae* and *Pseudomonas aeruginosa* in municipal wastewater by tetracationic porphyrin and violet-blue light: The impact of wastewater constituents

**Martina Mušković**[1☯], **Matej Planinić**[2☯], **Antonela Crepulja**[2], **Marko Lušić**[1], **Marin Glad**[3], **Martin Lončarić**[4], **Nela Malatesti**[1]*, **Ivana Gobin**[2]

1 Department of Biotechnology, University of Rijeka, Rijeka, Croatia, 2 Department of Microbiology and Parasitology, Faculty of Medicine, University of Rijeka, Rijeka, Croatia, 3 Department for Environmental Protection and Health Ecology, Teaching Institute of Public Health, Rijeka, Croatia, 4 Photonics and Quantum Optics Unit, Center of Excellence for Advanced Materials and Sensing Devices, Ruđer Bošković Institute, Zagreb, Croatia

☯ These authors contributed equally to this work.

* nela.malatesti@biotech.uniri.hr

## Abstract

There is an increasing need to discover effective methods for treating municipal wastewater and addressing the threat of multidrug-resistant (MDR) strains of bacteria spreading into the environment and drinking water. Photodynamic inactivation (PDI) that combines a photo-sensitiser and light in the presence of oxygen to generate singlet oxygen and other reactive species, which in turn react with a range of biomolecules, including the oxidation of bacterial genetic material, may be a way to stop the spread of antibiotic-resistant genes. The effect of 5,10,15,20-(pyridinium-3-yl)porphyrin tetrachloride (TMPyP3) without light, and after activation with violet-blue light (VBL) (394 nm; 20 mW/cm²), on MDR strains of *Pseudomonas aeruginosa*, *Klebsiella pneumoniae* and *K. pneumoniae* OXA-48 in tap water and municipal wastewater was investigated. High toxicity (~2 μM) of TMPyP3 was shown in the dark on both strains of *K. pneumoniae* in tap water, while on *P. aeruginosa* toxicity in the dark was low (50 μM) and the PDI effect was significant (1.562 μM). However, in wastewater, the toxicity of TMPyP3 without photoactivation was much lower (12.5–100 μM), and the PDI effect was significant for all three bacterial strains, already after 10 min of irradiation with VBL (1.562–6.25 μM). In the same concentrations, or even lower, an anti-adhesion effect was shown, suggesting the possibility of application in biofilm control. By studying the kinetics of photoinactivation, it was found that with 1,562 μM of TMPyP3 it is possible to achieve the complete destruction of all three bacteria after 60 min of irradiation with VBL. This study confirmed the importance of studying the impact of water constituents on the properties and PDI effect of the applied photosensitiser, as well as checking the sensitivity of targeted

**Data Availability Statement:** All relevant data are within the paper and its Supporting Information files.

**Funding:** Our study was financed by the University of Rijeka grants (UNIRI-INOVA to NM, and uniri-biomed-18-171 to IG) and the Ministry of Science and Education of Croatia (ERDF) grant for CEMS No. KK.01.1.1.01.0001. The funders had no role in study design, data collection and analysis, decision to publish, or preparation of the manuscript.

**Competing interests:** The authors have declared that no competing interests exist.

bacteria to light of a certain wavelength, in conditions as close as possible to those in the intended application, to adjust all parameters and perfect the method.

# Introduction

Photodynamic therapy (PDT) uses photoactive dye (= photosensitiser, PS), light and oxygen to create reactive oxygen species (ROS), most importantly singlet oxygen ($^1O_2$), to induce a cytotoxic effect. Although the PDT effect was discovered on the microorganism (*Paramecium*), PDT has been developed primarily for applications in antitumor therapies [1]. In recent years, antimicrobial PDT (aPDT), known also as photodynamic inactivation (PDI) and photodynamic antimicrobial chemotherapy (PACT), has been increasingly developed to treat various (usually local) infections, and for disinfection purposes, such as the disinfection of water, blood, (hospital) surfaces and medical devices, food, and crops [2]. New antimicrobial approaches, such as aPDT, are needed because the excessive and inappropriate use of antibiotics has led to the emergence of antibiotic resistance (ABR), which has been declared by the World Health Organization (WHO) as one of the greatest dangers to global health and development [3]. Shortly thereafter, WHO declared this kind of danger for antimicrobial resistance (AMR) in general, which includes multidrug resistance (MDR) [4]. Among the most notorious pathogens that pose a particularly significant health threat due to their MDR and an elevated risk to cause nosocomial or healthcare-associated infections (HAI), are ESKAPE bacteria and biofilms they produce. ESKAPE is an acronym that stands for the names of Gram-positive bacteria, *Enterococcus faecium* and *Staphylococcus aureus*, and Gram-negative, *Klebsiella pneumoniae*, *Acinetobacter baumannii*, *Pseudomonas aeruginosa* and *Enterobacter species* [5], and at the same time connects their high tendency to "escape" from the bactericidal activity of many antibiotics used in clinics [6]. Municipal wastewater from public sewage contains stormwater and wastewater from households, industries, and hospitals, and can be burdened by MDR bacterial strains. A recent study of facultative pathogenic bacteria at twenty-three wastewater treatment plants in Germany found a link between clinically relevant antibiotic resistance genes (ARGs) and hospital wastewaters [7]. Not only are ARGs present in municipal and hospital waters, but it has been shown that wastewater treatment plants in urban watersheds have a role in their spread, therefore AMR and MDR into the environment [8].

There are usually four levels of wastewater treatment, which include preliminary and primary treatments that aim at removing coarse and settleable suspended solids respectively, by mostly physical mechanisms, secondary treatment that aims at removing organic matter by (mostly) biological mechanisms, and tertiary treatment, which is intended to remove toxic and other pollutants, and pathogens [9]. The treatment of wastewater from urban areas consists of pre-treatment, primary and secondary treatment before being released into natural waters, and sometimes a tertiary treatment is applied [10]. Water disinfection is the last stage and the only obligatory part of tertiary treatment and can be used to combat ARGs and their spread through horizontal gene transfer. The primary chemical oxidants currently used for water disinfection include free chlorine, free bromine, chlorine dioxide ($ClO_2$) and ozone. A recent review and comparison of their mechanisms of action has revealed that the strongest disinfectant is $ClO_2$, while ozone has the highest average inactivation [11]. All these primary oxidants react mostly with certain amino acid side groups, thus they can damage intracellular and membrane proteins; ozone in addition can react with double bonds in lipids and can damage membranes more substantially, while $ClO_2$ is the most selective and reacts only with thiols, phenols, and aromatic amines [11]. Secondary oxidants, such as the hydroxyl radical formed from the

ozone primary reactions, are more likely than primary oxidants to react with nucleic acids and oxidise genetic material, which can lead to deactivation of the ARGs [11]. However, there are certain deficiencies associated with these and other disinfectants currently being applied, such as the formation of toxic by-products (especially with chlorination), toxicity to aquatic animals and plants (ozonation and ultraviolet (UV) irradiation), being expensive (ozonation) and possibly mutagenic (UV) [10]. Moreover, current wastewater treatments have proven ineffective against certain resistant bacterial strains, and in some cases are even considered responsible for increasing their resistance. For example, a recent study showed that chlorine disinfection increased the proportion of extended-spectrum beta-lactamase (ESBL)-producing bacteria in hospital wastewater [12]. Therefore, it is necessary to explore other ways of disinfecting wastewater to control MDR pathogenic bacteria.

Singlet oxygen ($^1O_2$) and other ROS generated through Type II and Type I processes in PDT respectively, react with biomolecules to which the PS is close enough, and which may include lipids, amino acids, and proteins, as well as nucleic acids, so they can oxidise genetic material, which can lead to the deactivation of the ARGs. Photodynamic therapy can be used against bacteria in planktonic state as well as in biofilm, but also against other pathogenic microbes, and resistance is not likely to develop due to multitarget mechanism [2]. Moreover, it could be used for the photodegradation of pollutants in wastewater such as phenol [13]. However, the potential of aPDT has long been neglected, and only recently, amid the great need for new antimicrobial methods, therapies, and agents, it is beginning to receive increased attention [14]. PDI has been shown effective against resistant strains, and even more in combination with antibiotics thanks to MDR efflux inhibition [15], so antibiotics present in hospital wastewater could enhance PDI and help fight against MDR [16]. PDI could therefore be a suitable alternative to chemical disinfection methods, or could be used in combination with them, depending on the following factors. When using PDI for water disinfection, the type of PS has to be taken into an account (with known photophysical properties, $^1O_2$ and ROS production, photostability, 'dark toxicity'), and its concentration, the applied light wavelength and fluence rate, water pH and water quality (e.g. turbidity negatively affects PDI), whether the PS is free or immobilised (through adsorption, electrostatic interaction, conjugation etc.), and on what material (e.g. glass, resins, polymers) [17]. Photosensitisers immobilised on solid support may be used for water disinfection repeatedly and simply removed after use, recycled, and for photoactivation both artificial light and solar energy can be employed, which makes PDI economical and safe treatment for the environment [18, 19]. Free PSs may decompose under (sun)light, or they can be removed from water by adsorption on activated carbon filters, same as their photoproducts [20].

Porphyrins are one of the most widely used groups of PSs for PDT and are also very suitable for various applications in water and water treatments. Their absorption is throughout the visible part of the electromagnetic (EM) spectrum, so they can be activated by artificial light as well as sunlight. Their particularly high absorption at the Soret band corresponds to blue light, which is the wavelength that penetrates the deepest in water. Furthermore, water soluble cationic porphyrins, especially those with higher number of positive charges, proved superior to neutral and anionic porphyrins for PDI in wastewater against both Gram-positive (*S. aureus* and *Enterococcus*) and Gram-negative (*E. coli*) sewage bacteria as well as sewage T4-like bacteriophage [10, 18, 21–24]. The advantage of using cationic porphyrins over neutral and anionic in the photoinactivation of various pathogens, and particularly Gram-negative bacteria has been repeatedly proven, and the latter is explained by the formation of electrostatic interactions between a PS and the cell wall that improves the PDI response [25, 26]. Furthermore, tri- and tetracationic pyridinium porphyrins seem to have stood out as promising for a variety of possible PDI applications [27], including photoinactivation of bacteria in wastewater [28].

**Fig 1. Structure of the tested porphyrin (TMPyP3), and porphyrins TMPyP3-CH₃, TMPyP3-C₁₇H₃₅ and TMPyP4.**

Although 5,10,15,20-(pyridinium-3-yl)porphyrin tetrachloride (TMPyP3) and its more researched regioisomer 5,10,15,20-(pyridinium-4-yl)porphyrin tetrachloride (TMPyP4) (Fig 1), are both water soluble and excellent $^1O_2$ producers, in some reports TMPyP4 has been shown to be less stable than other tested PSs, and less PDI efficient than TMPyP3 [15, 29]. In our previous work we examined the PDT effect and inactivation potential of tetracationic symmetric TMPyP3, and two asymmetric tricationic pyridinium porphyrins, one hydrophilic (TMPyP3-CH₃) and one amphiphilic with a long alkyl chain (TMPyP3-C₁₇H₃₅) (Fig 1), against *Legionella pneumophila*, an environmental Gram-negative bacterium that is difficult to treat with currently used water disinfection methods [30, 31]. All tested porphyrins were effective against *Legionella* and its biofilm, among which amphiphilic TMPyP3-C₁₇H₃₅ with the lowest minimal effective concentration (MEC) values, however, while two hydrophilic porphyrins were stable in all water samples, the amphiphilic one was prone to aggregation and shown unstable in hard and soft water samples [31]. Between the two hydrophilic porphyrins, TMPyP3 showed a stronger antimicrobial effect, possibly due to a higher number of positive charges (4 *vs* 3 in TMPyP3-CH₃), and given that it is also synthetically more accessible, and therefore more economically acceptable for applications such as water disinfection, we decided to use it in this study.

In a recent review (2023) on the impact of water constituents on the photoinactivation of bacteria, it was pointed out that of the nearly 11,000 publications describing inactivation of bacteria by different photo-disinfection processes in water, only the 4.4% of them described water matrix and reported the effect of water constituents [32]. While turbidity is known to have a negative effect due to light attenuation, other water constituents, such as dissolved

organic matter (DOM), can have different, both negative and positive effects [32]. Therefore, one of the aims of this study was to investigate the influence of wastewater constituents on the properties of the used exogenous PS (TMPyP3) and PDI effect. Given that in our previous work we analysed the impact of inorganic ions present in tap water on TMPyP3 and its PDI effect [30, 31], and since tap water is part of the municipal wastewater, in this work we use tap water for comparison to emphasize the impact of additional, mainly organic, matter present in wastewater.

Among ESKAPE bacteria, *P. aeruginosa* and *K. pneumoniae* are both Gram-negative, rod-shaped, opportunistic pathogens found in water and soil, responsible for a considerable proportion of HAI, and amongst those found by WHO as critical, i.e., the highest on the global priority list of antibiotic-resistant bacteria that require innovative approaches [33]. They are also interesting for their different susceptibility to violet-blue light, which is due to differences in the content of endogenous PSs [34]. Unlike *P. aeruginosa*, *K. pneumoniae* is still underrepresented in the literature in the context of photodynamic inactivation (PDI), and most of these publications deal with the treatment of infections, while PDI using an exogenous porphyrin on *K. pneumoniae* in wastewater has so far not been described. Therefore, our aim was to study PDI using TMPyP3 as an exogenous PS, with violet-blue light for its photoactivation, against MDR strains of *P. aeruginosa* and *K. pneumoniae* in municipal wastewater. Minimal effective concentration (MEC) and minimal anti-adhesion concentration (MAAC) values of TMPyP3 in tap water and municipal wastewater were obtained, and PDI with different TMPyP3 concentrations and doses of violet-blue light (394 nm) were analysed.

## Materials and methods

### Photosensitiser (PS)

The porphyrin used as the PS in this work, 5,10,15,20-(pyridinium-3-yl)porphyrin tetrachloride (TMPyP3) (Fig 1) was synthesised as previously described [30], and its structure was confirmed by $^1$H NMR spectroscopy:

$^1$H NMR (MeOH-d4): δ/ppm 4.80 (s, 12H, 4 x N-C$H_3$), 8.52–8.58 (m, 4H, Py-5-$H$), 9.05 (br s, 8H, β-$H$), 9.39 (d, 4H, $J$ = 6.6 Hz, Py-6-$H$), 9.47 (d, 4H, $J$ = 6.4 Hz, Py-4-$H$), 9.96 (s, 4H, Py-2-$H$).

A stock solution of TMPyP3 (200 μM) for microbiological experiments was prepared in sterile tap water (sterilised by autoclaving at 121˚C, 1.2 bar, 20 min). After dissolving porphyrin in water, the solution was filtered using sterile Syringe Filter 0,45 μm (Labex Ltd, Budapest, Hungary), and kept covered in the dark at 4˚C until use.

### Spectroscopic properties of TMPyP3

Absorbance spectra were obtained on the Cary 60 UV-Vis spectrophotometer and fluorescence spectra on the Cary Eclipse fluorescence Spectrophotometer, both from Agilent Technologies (Santa Clara, California, USA). Two concentrations of TMPyP3 for spectroscopic analysis (10 μM for absorption and 1 μM for fluorescence spectra) were prepared in municipal wastewater. Both absorbance and fluorescence spectra were recorded in a 1 cm fluorescence quartz cuvette (Hellma Analytics, Müllheim, Germany). For obtaining fluorescence spectra, the Soret band wavelength ($\lambda$ = 423 nm) was used for the excitation.

### (Photo)stability of TMPyP3

The stability of TMPyP3 in wastewater was tested as previously described [31]. Shortly, the decrease of absorbance of the porphyrin solution in municipal wastewater (10 μM) at the Soret

band wavelength ($\lambda$ = 420 nm) was measured. In the first experiment (T1), a TMPyP3 solution was irradiated with violet-blue light for 10 min ($\lambda$ = 411 nm, fluence rate 11 mW/cm$^2$, total light dose 6.6 J/cm$^2$) on the first day, and its stability was measured over 5 days, while in the second experiment (T2), the solution was irradiated every day for 10 min (total light dose 33 J/cm$^2$) and the absorbance was measured after the treatment. Between the measurements, the solutions were kept in the dark at room temperature (25˚C). As a 'dark control', the solution was kept in the dark under the same conditions for 5 days without irradiation, and the absorbance was measured every 24 hours.

## Singlet oxygen ($^1O_2$) production

The production of singlet oxygen and other ROS was evaluated by measuring the decrease of fluorescence of commercially available fluorescent dye, 1,3-diphenylisobenzofuran (DPBF) (Sigma Aldrich, St. Louis, MO, USA). This dye reacts with $^1O_2$ yielding non-fluorescent 1,2-dibenzoyl benzene (DBB), so the fluorescence decrease of DPBF, which is a result of a loss of the extended $\pi$-electron system (due to the production of DBB), is proportional to the trapped $^1O_2$ produced by TMPyP3 [35]. The porphyrin TMPyP4 was synthesised as previously described [36], and used as a reference for comparison with TMPyP3. Stock solutions of DPBF and for both porphyrins were prepared in dimethylsulfoxide, DMSO (Sigma Aldrich, St. Louis, MO, USA) (10 mM). The control dye DPBF was further diluted in ethanol (VWR, Radnor, Pennsylvania, USA) and the porphyrins in different water samples (DEMI water, tap water and wastewater) until the final concentrations (8 μM for DPBF in ethanol, and 1 μM for the porphyrins in all water samples). The solutions were mixed at 1:1 ratio and irradiated for 10 minutes with violet-blue light ($\lambda$ = 411 nm, 3.5 mW/cm$^2$, total light dose 2.1 J/cm$^2$) at room temperature and under constant stirring. The fluorescence decrease was measured at $\lambda_{emm}$ = 453 nm ($\lambda_{ex}$ = 420 nm) at the beginning of the experiment and every 60 s. The results are shown as the mean of three measurements with standard deviation (SD) of the area under the curve (AUC) calculated from the normalised DPBF fluorescence decay, according to the formula:

$$AUC = \frac{\left(\left(\frac{I}{I_0}\right)s + \left(\frac{I}{I_0}\right)f\right)}{2(t_f - t_s)} \tag{1}$$

$(I/I_0)_s$—fluorescence intensity and initial fluorescence intensity ratio at the beginning of the 60 s interval;

$(I/I_0)_f$—fluorescence intensity and initial fluorescence intensity ratio at the end of the 60 s interval;

$t_s$—time at the beginning of the 60 s interval;

$t_f$—time at the end of the 60 s interval.

In addition, the kinetic results in $^1O_2$ production were calculated by linearizing the given curves of fluorescence decrease using the ln function (S2 Dataset for DEMI water and wastewater, and S3 Dataset for the measurements in tap water).

## Light sources

Light sources were designed and made in the Laboratory for Photonics and Quantum Optics at Ruđer Bošković Institute (Zagreb, Croatia). For the photostability and singlet oxygen production, measurements LED-based source of violet-blue light with adjustable fluence rate was used ($\lambda$ = 411 nm, $\Delta\lambda_{FWHM}$ = 20 nm; fluence rates 3.5 and 11 mW/cm$^2$), and PDI studies were conducted using a LED-based source of violet-blue light with higher fluence rate, set by the

distance from the microtiter plate ($\lambda$ = 394 nm, $\Delta\lambda_{FWHM}$ = 14 nm; fluence rate 20 mW/cm$^2$), as in our previous work [31].

## Water-sampling and physicochemical analyses

Municipal wastewater samples were obtained from, and its physicochemical properties determined, by the Teaching Institute of Public Health of Primorje-Gorski Kotar county (S1 Table). The Teaching Institute is authorised by the Ministry of Environmental Protection and Energy of the Republic of Croatia and accredited by the Croatian Accreditation Agency for wastewater sampling and analysis. The wastewater used in this study was sampled from a wastewater treatment plant Delta (city of Rijeka, Croatia). The samples for the purposes of this study were collected during regular monthly sampling. Before the experiments, wastewater was filtered using the Whatman No. 3 filter paper (Macherey-Nagel, Duren, Germany), and sterilised by autoclaving (121˚C, 1.2 bar, 20 min). Physicochemical properties of sampled wastewater were determined before, and after filtration and sterilisation (S1 Table). Samples of DEMI and tap water for all experiments were sterilised using the same procedure as described above, and all water samples were kept at 4˚C until use.

## Bacterial strains and sample preparation

Clinical isolates of the following bacterial strains were used in this work: *Klebsiella pneumoniae* (ATCC 700603), *Klebsiella pneumoniae* strain expressing OXA-48 (NCTC 13442), and *Pseudomonas aeruginosa* (ATCC 27853). The antibiotic susceptibility tests were made according to the European Committee for Antimicrobial Susceptibility Testing (EUCAST) recommendations (The European Committee on Antimicrobial Susceptibility Testing, Version 13.0, 2023). The results of the antibiotic sensitivity test performed for these strains are given in the S2 Table. Bacteria were inoculated on Mueller-Hinton agar and incubated for 24 h at 35±2˚C prior to preparation of the concentration needed for the experiments. An initial bacterial suspension was prepared by resuspending bacteria from the agar in 0.8% saline, and the bacterial concentration was determined by measuring optical density (OD). The initial bacterial suspension was approximately 10$^8$ CFU/ mL, while in the experiments the initial concentration was diluted to 10$^6$ CFU/mL in the tested water samples (tap water or municipal wastewater).

## Survival of multi-resistant bacteria in different water samples

The bacterial suspension (10$^6$ CFU/mL) was diluted in tap water or municipal wastewater in a ratio of 1:9. Both initial water samples (tap water or municipal wastewater) were plated on Mueller-Hinton agar in multiple dilutions and incubated for 24 h at 35±2˚C to determine the number of bacteria. The same procedure was repeated on days 3, 5, 7, 9 and 14 days after the preparation of the initial sample to determine the bacterial survival over 14 days. Between the experiments, the initial samples were kept at 30˚C. All measurements were performed in triplicate (*P. aeruginosa* survival in quadruplicate) and values are presented as an average of multiple measurements with standard deviation (SD values given in S4 Dataset).

## Determination of minimal effective concentration (MEC) and minimal anti-adhesive concentration (MAAC)

The bacterial suspension was prepared according to the protocol described above, in concentration 10$^6$ CFU/mL in DEMI water, tap water and municipal wastewater. Porphyrin (50 μL) was added to the microtiter plate in different concentrations. The bacterial suspension was

added (50 μL) and incubated at room temperature for 30 min under constant shaking. After the incubation, the plates were irradiated with violet-blue light for 10 min ($\lambda = 394$ nm; fluence rate 20 mW/cm$^2$; total light dose 12 J/cm$^2$), followed by incubation at 35±2˚C. After 24 h incubation in the dark, 10 μL of the treated suspension was added in duplicate on Mueller-Hinton agar, followed by cultivation of the bacteria for another 24 h at 35±2˚C. A 'dark control' was prepared using the same protocol, but without the irradiation step.

To determine the minimal anti-adhesive concentration (MAAC), a similar protocol with the following additional steps was used. After a 24-h dark incubation, the plates were washed twice with tap water to remove all bacteria that did not attach to the surface. The samples were then sonicated for 1 min and resuspended in sterile tap water before being plated on a Mueller-Hinton agar for the next 24 h of incubation.

The MEC value was determined as the lowest concentration of the porphyrin that reduces bacterial growth by 99.9%, and the MAAC value as the lowest concentration that affects the bacterial adhesion to the surface of polystyrene. All the experiments to obtain the MEC and MAAC values were repeated, and the same values as in the first experiment were also obtained in the second.

## Photoinactivation (PDI) kinetics of different bacterial strains with TMPyP3 in wastewater

Different water samples were prepared with TMPyP3 in concentrations 0.5 × MEC, 1 × MEC and 2 × MEC values. The prepared concentrations were mixed with bacterial suspension at the ratio of 1:1 (50 μL of porphyrin solution and 50 μL of bacterial suspension) and incubated for 30 min at room temperature while constantly shaking. The samples were irradiated with violet-blue light ($\lambda = 394$ nm; 20 mW/cm$^2$) for 15 min (total light dose 18 J/cm$^2$), 30 min (36 J/cm$^2$) and 60 min (72 J/cm$^2$). After the irradiation, the samples were plated on a Mueller-Hinton agar and incubated for 24 h at 35±2˚C. All measurements were performed in a minimum of five replications, and the results are presented as mean with SD as error bars (SD values given in S4 Dataset).

## Scanning transition electron microscopy (STEM) of *P. aeruginosa* and *K. pneumoniae* OXA-48

A bacterial suspension ($10^8$ CFU/mL) of *P. aeruginosa* and *K. pneumoniae* OXA-48 in tap water was treated with 1 × MEC concentration of TMPyP3 (1.562 and 3.125 μM for *P. aeruginosa* and *K. pneumoniae*, respectively). After a 30-min incubation, the samples were irradiated with violet-blue light for 10 min ($\lambda = 394$ nm; 20 mW/cm$^2$; 12 J/cm$^2$). After the treatment, 10 μL of the treated suspension was added to Formvar-coated copper grids (Agar Scientific Ltd.) and left for 2 min. Prior and after staining with 1% phosphotungstic acid (PTA, Sigma-Aldrich) for 1 min, the excess of liquid was removed from the grids using the Whatman No. 3 filter paper (Macherey-Nagel). For the scanning-transition electron microscopy (STEM) analysis, Jeol JSM-7800F field scanning microscope equipped with STEM detector was used and prepared grids were ready to use after air drying for 20 h. All pictures were taken at a 15k magnification.

## Statistical analyses

All statistical analyses were performed using GraphPad Prism 8.0.1. The results are shown as an average of the repeated measurements with SD used as an error range. In the experiment of (photo)stability the results were compared using two-way ANOVA analysis, while in the

experiments of singlet oxygen detection and in PDI studies, an unpaired student t-test was performed ($p < 0.05$).

## Results

### Photostability and singlet oxygen production of TMPyP3 in municipal wastewater

The physical characterization of the sample of municipal wastewater taken for this study included measuring temperature, flow, pH, conductivity, and visible properties of the water, while the chemical characterization included measuring the amount of minerals and organic matter present in water (S1 Table). The wastewater sample was filtered (using Whatman No. 3 filter paper) to remove only larger pieces of waste, while the dissolved organic matter (DOM) can pass through the filter paper used, so it remains in the sample. This sample was then sterilized by autoclaving, and, considering that autoclaving can change also the physicochemical properties of wastewater, physicochemical analyses were carried out both before and after autoclaving (S1 Table). Most of the measured parameters have lower values in autoclaved wastewater, however, there is still a significant amount of organic material (COD = 189 mg/L per $O_2$), chloride (6073 mg/L), and other inorganic matter present (S1 Table).

The photophysical and photochemical properties of TMPyP3 in the wastewater sample were evaluated by absorption and fluorescence spectroscopy, and (photo)stability, and singlet oxygen production ($^1O_2$) were measured to examine the influence of the inorganic and organic matter contained in water. In the absorption spectra, a strong peak belonging to the Soret band is visible at 423 nm, and four Q bands between 500 and 650 nm. Two of the Q bands, at 570 and 651 nm are of low intensity and hardly detectable in the spectra. Two peaks at 658 and 706 nm visible in the fluorescence spectrum are of very low intensity (Fig 2).

The (photo)stability of TMPyP3 in municipal wastewater was monitored in a five-day experiment, similar to our previous work [31], and the porphyrin solution was either

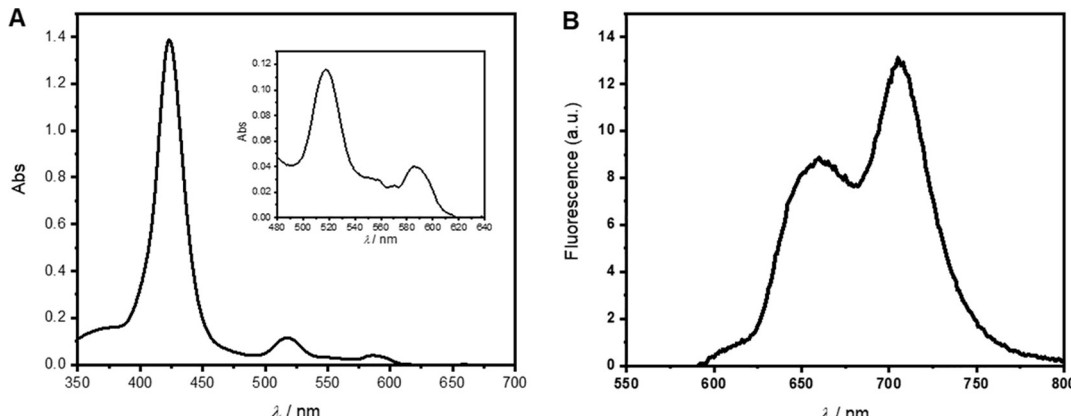

| Absorbance: λ/nm ($\varepsilon \times 10^3$ M$^{-1}$cm$^{-1}$) | | | | | Fluorescence: λ/nm | |
|---|---|---|---|---|---|---|
| Soret (B) | Qy (1-0) | Qy (0-0) | Qx (1-0) | Qx (0-0) | Q (1-0) | Q(0-0) |
| 423 (148.4) | 517 (14.6) | 570 (5.0) | 586 (6.5) | 651 (1.8) | 658 | 706 |

**Fig 2. Spectral characteristics of porphyrin TMPyP3 in municipal wastewater.** Absorption (A) and fluorescence (B) spectra of TMPyP3 (10 μM for absorption and 1 μM concentration for fluorescence spectra) in municipal wastewater, with absorption and emission bands, and molar absorption coefficients in parentheses (table).

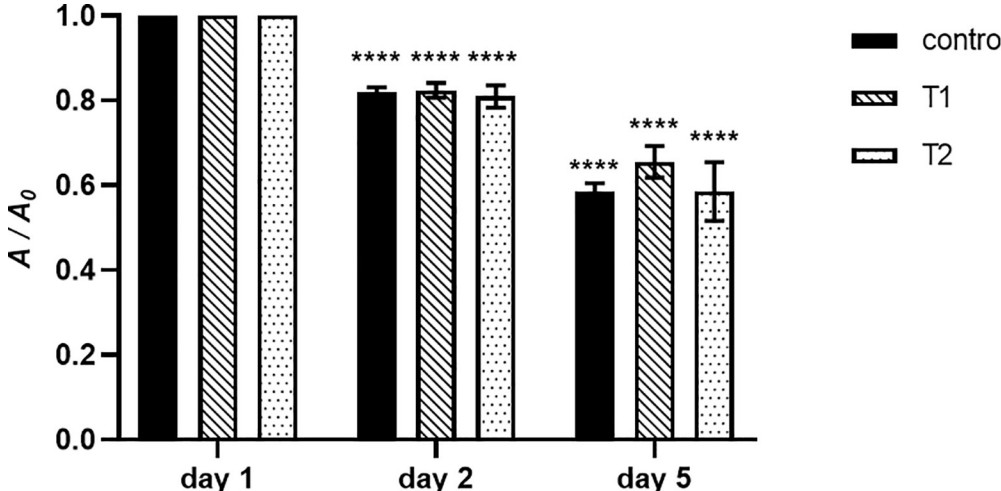

**Fig 3. Photostability of TMPyP3 in municipal wastewater on the 2nd and 5th day.** The stability of TMPyP3 in water samples was investigated over 5 days by measuring absorbance at the Soret band ($\lambda$ = 423 nm) of the 10 μM porphyrin solution. Absorbance was measured after one exposure to violet-blue light ($\lambda$ = 411 nm; 11 mW/cm$^2$) for 10 min (T1) (total light dose 6.6 J/cm$^2$), and after multiple exposures (every 24 h) to the same irradiation conditions for 10 min (T2) (total light dose 33 J/cm$^2$). The results are presented as a measurement average of triplicate measurements ± SD; **** $p < 0.0001$, *** $0.001 > p > 0.0001$, ** $0.1 > p > 0.001$, * $p > 0.1$, ns $p > 0.5$.

irradiated with violet-blue light once on the first day (T1), or every 24 hours (T2), or it was not exposed to light at all (control), and the absorbance (*A*) measured on each day was compared with the initial absorbance ($A_0$). After the first 24 hours, according to this difference ($A / A_0$), TMPyP3 stability decreased ~20% in all three measurements (control, T1 and T2), and after five days, 60% of the $A_0$ remained in all three cases (Fig 3). No significant difference was observed among the three experiments on the 2nd and 5th day. The measured stability during all five days showed that after a 20% decline in stability after the first day, other days the stability decreases less and at about 10% each day (S1 Dataset).

The singlet oxygen (and other ROS) production of TMPyP3 was determined by measuring the decrease of fluorescence intensity of 1,3-diphenylisobenzofuran (DPBF) in municipal wastewater and DEMI water at $\lambda$ = 453 nm every 60 s for 10 minutes (S2 Dataset). TMPyP4 was used as a reference compound, and it was measured under the same conditions as TMPyP3. A measured decrease in the fluorescence intensity of DPBF under the same conditions without the addition of TMPyP3 was used as a control. A reduction of 60% in DEMI water and ~65% in wastewater was detected for DPBF control (Fig 4). However, a ~90% of DPBF fluorescence decrease was observed in the presence of TMPyP3 in both water samples, indicating $^1O_2$ production by TMPyP3. Compared to TMPyP4, TMPyP3 shows better $^1O_2$ production in DEMI water, and in municipal wastewater (Fig 4), also in tap water, although with smaller difference (S3 Dataset). A slightly stronger decrease in fluorescence intensity in the DPBF control in wastewater (Fig 4B) compared to the decrease in DEMI water (Fig 4A), indicates that, relative to the DPBF control, there is also somewhat lower $^1O_2$ production by TMPyP3 in wastewater compared to DEMI water (Fig 4A and 4B).

## Minimal effective (MEC) and minimal anti-adhesion (MAAC) concentrations of TMPyP3 in tap water and municipal wastewater

To study the antibacterial and PDI activity of TMPyP3 against *P. aeruginosa*, *K. pneumoniae* and *K. pneumoniae* OXA-48 in municipal wastewater, the minimal effective concentration

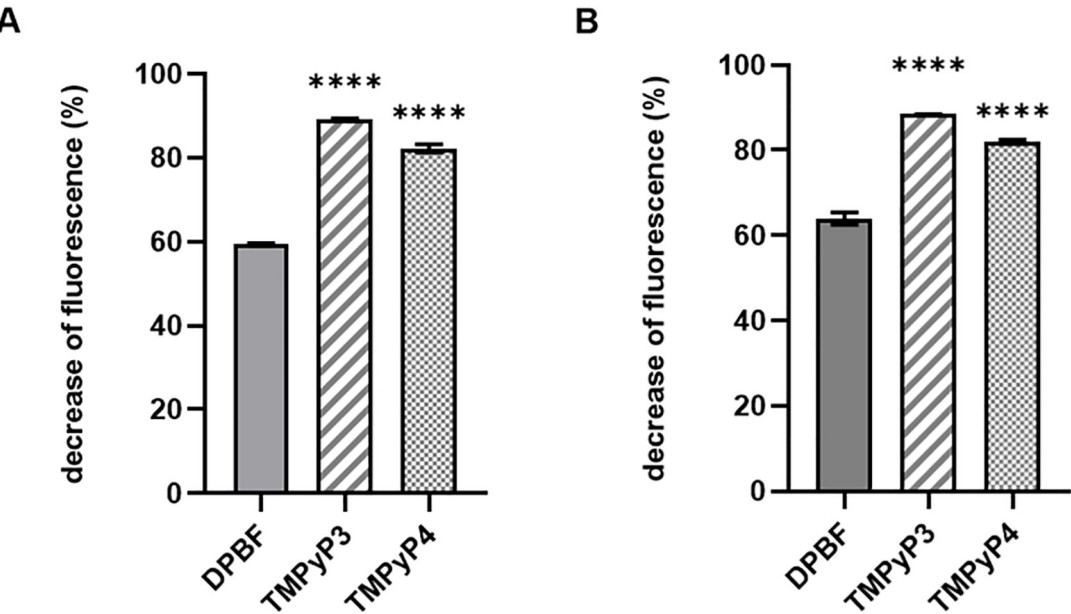

**Fig 4.** Singlet oxygen production of TMPyP3 in DEMI water (A) and municipal wastewater (B). The singlet oxygen production was evaluated as the decrease (%) of fluorescence intensity of 1,3-diphenylisobenzofuran (DPBF) (4 µM) at $\lambda$ = 453 nm in 50% EtOH and 50% DEMI water (A) or municipal wastewater (B) solution with porphyrin TMPyP3 (1 µM). The porphyrin TMPyP4 was used as a reference and tested under the same conditions as TMPyP3. The control is the decrease of fluorescence intensity of DPBF measured under the same conditions, but without TMPyP3 in the water sample. Results are shown as an average of three measurements with SD as error bars: **** $p < 0.0001$, *** $0.001 > p > 0.0001$, ** $0.1 > p > 0.001$, * $p > 0.1$, ns $p > 0.5$.

(MEC) (Table 1) and the minimal anti-adhesive concentration (MAAC) values (Table 2) were determined after 10-minute violet-blue irradiation (irr. $\lambda$ = 394 nm; 20 mW/cm$^2$), and without any exposure to light (DARK) in tap and municipal wastewater. In addition, a 30-minute violet-blue irradiation was performed in municipal wastewater to investigate the impact of the dose of light on the MEC (Table 1B, the last column).

Low MEC values (1.562–6.125 µM) were determined after irradiation for 10 min on all bacterial strains in both types of water. However, on *K. pneumoniae* strains, a strong dark toxicity of TMPyP3 (3.125 µM and 1.562 µM for *K. pneumoniae* and *K. pneumoniae* OXA-48, respectively) was observed in tap water, as opposed to the low dark toxicity for *P. aeruginosa* (50 µM) (Table 1A). A significantly lower dark toxicity (25–100 µM) was observed for all bacterial

**Table 1. Minimal effective concentration (MEC) values of TMPyP3 in tap water (A) and municipal wastewater (B).**

| A | DARK | 10 min irr. $\lambda$ = 394 nm; 20 mW/cm$^2$ |
|---|---|---|
| *P. aeruginosa* | 50 µM | 1.562 µM |
| *K. pneumoniae* | 3.125 µM | 1.562 µM |
| *K. pneumoniae* OXA-48 | 1.562 µM | 3.125 µM |

| B | DARK | irr. $\lambda$ = 394 nm; 20 mW/cm$^2$ | |
|---|---|---|---|
| | | 10 min | 30 min |
| | | dose of light 12 J/cm$^2$ | dose of light 36 J/ cm$^2$ |
| *P. aeruginosa* | 100 µM | 6.25 µM | 3.125 µM |
| *K. pneumoniae* | 50 µM | 1.562 µM | 1.562 µM |
| *K. pneumoniae* OXA-48 | 25 µM | 3.125 µM | 1.562 µM |

**Table 2. Minimal anti-adhesion concentration (MAAC) values of TMPyP3 in tap water (A) and municipal wastewater (B).**

| A | DARK | 10 min irr. $\lambda$ = 394 nm; 20 mW/cm$^2$ |
|---|---|---|
| *P. aeruginosa* | 50 μM | 1.562 μM |
| *K. pneumoniae* | 3.125 μM | 0.78 μM |
| *K. pneumoniae* OXA-48 | 1.562 μM | 3.125 μM |
| **B** | **DARK** | **10 min irr. $\lambda$ = 394 nm; 20 mW/cm$^2$** |
| *P. aeruginosa* | 100 μM | 6.25 μM |
| *K. pneumoniae* | 25 μM | 1.562 μM |
| *K. pneumoniae* OXA-48 | 12.5 μM | 3.125 μM |

strains in wastewater, but again the lowest on *P. aeruginosa* and the highest for *K. pneumoniae* OXA-48 (Table 1B). The highest phototoxic index, i.e., the difference between the MEC values determined without and with irradiation (MEC$_{DARK}$ / MEC$_{irr}$), was obtained on *P. aeruginosa* in tap water after 10 min (Table 1A), and in wastewater after 30 min of irradiation, and on *K. pneumoniae* in wastewater after 10 min of irradiation. The same MEC value and phototoxic index (= 32) however, remained for *K. pneumoniae* after 30 min of irradiation in wastewater (Table 1B). A very small phototoxic index (= 2) is calculated from the MEC values obtained on *K. pneumoniae* in tap water, while for *K. pneumoniae* OXA-48 the MEC value after 10-min irradiation (3.125 μM) was even higher than MEC indicating strong dark toxicity (1.562 μM) (Table 1A). The MEC values were reduced in half as a result of longer irradiation and a higher light dose in wastewater for *P. aeruginosa* and *K. pneumoniae* OXA-48, while, as mentioned above, this had no impact on *K. pneumoniae* (Table 1B). For comparison, the MEC values for TMPyP4 were determined on all three bacterial strains in wastewater without light (DARK), and after irradiation ($\lambda$ = 394 nm; 20 mW/cm$^2$) for 10 min. The obtained MEC values were slightly higher (S3 Table) than for TMPyP3, in accordance with the measured $^1O_2$ production (Fig 4).

To investigate the potential of TMPyP3 and PDI against biofilm formation in municipal wastewater, the MAAC values were obtained, indicating the concentration of TMPyP3 that inhibits the adhesion of the bacteria to the surface, in tap and wastewater, without (DARK) or with 10-min irradiation. The MAAC values determined in tap water, with and without irradiation (Table 2A), are the same as respective MEC values (Table 1A), except for *K. pneumoniae* whose MAAC upon irradiation is half lower (0.78 μM) (Table 2A). Similarly, the MAAC values determined in wastewater after 10 min of irradiation (Table 2B), are the same for all bacterial strains as their respective MEC values (Table 1B), and so is the MAAC value for *P. aeruginosa* without irradiation (DARK in Tables 1B and 2B). However, the MAAC values for *K. pneumoniae* and *K. pneumoniae* OXA-48 obtained in wastewater without exposure to light are half lower (25 and 12.5 μM) (Table 2B) than their respective MEC values (Table 1B).

## Photoinactivation of *K. pneumoniae*, *K. pneumoniae* OXA-48 and *P. aeruginosa* in tap water and municipal wastewater

Before studying the bacterial (photo)inactivation kinetics with TMPyP3 in municipal wastewater, the bacterial survival in tap and wastewater was monitored for 14 days. *P. aeruginosa* is a Gram-negative ubiquitous microorganism capable of surviving under a variety of extreme environmental conditions and it is known that it degrades a broad range of organic molecules for nutrition to survive for a long period of time. In our study, *P. aeruginosa* easily survived in both tap water and municipal wastewater over 14 days (Fig 5A). In both water samples, a growth (1 log$_{10}$ CFU/mL) was observed on day 1, followed by a decline that was detected on

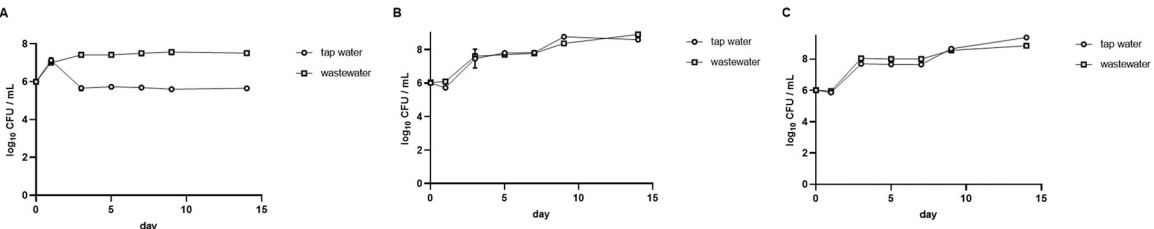

**Fig 5.** Survival of *P. aeruginosa* (A), *K. pneumoniae* (B) and *K. pneumoniae* OXA-48 (C) in sterile tap water and sterile municipal wastewater. After the preparation of the initial bacterial suspension $10^6$ CFU/ mL in wastewater or in tap water, the samples were kept at $35\pm2°C$ and colony-forming units were counted on days 1, 3, 5, 7, 9 and 14. The results are shown as an average of multiple measurements ± SD.

day 3 in wastewater (~2 $\log_{10}$ CFU/mL), while no changes in *P. aeruginosa* CFU number were observed in the tap water sample over 14 days. After the third day, there were no further changes in CFU/mL number until the end of the experiment in the wastewater sample (day 14) (Fig 5A).

For both *K. pneumoniae* strains the number of CFU increased with the same pattern of growth over 14 days of the experiment (Fig 5B and 5C). A significant rise in the growth curve of almost 2 $\log_{10}$ CFU/mL was detected on the third day, followed by stagnation until the second, smaller rise on the ninth day. For both strains, the number of colonies continued to increase until the end of the experiment in wastewater, and in tap water with stagnation for *K. pneumoniae* (Fig 5B) and slower growth for *K. pneumoniae* OXA-48 (Fig 5C). The total increase of growth of *K. pneumoniae* and *K. pneumoniae* OXA-48 over 14 days was almost 3 $\log_{10}$ CFU/mL.

Inactivation kinetics experiments with different concentrations of TMPyP3 were performed on all three bacterial strains, in tap water and municipal wastewater, first without irradiation to measure 'dark toxicity' for 60 min (Fig 6). *P. aeruginosa* showed the highest resistance to TMPyP3 without illumination, and a ~2 log decrease was observed with 50 μM and 100 μM of TMPyP3 in tap water (Fig 6A). In wastewater, almost no decrease was observed in both concentrations up to 60 min, and only a ~1 $\log_{10}$ CFU/mL reduction was detected at the end

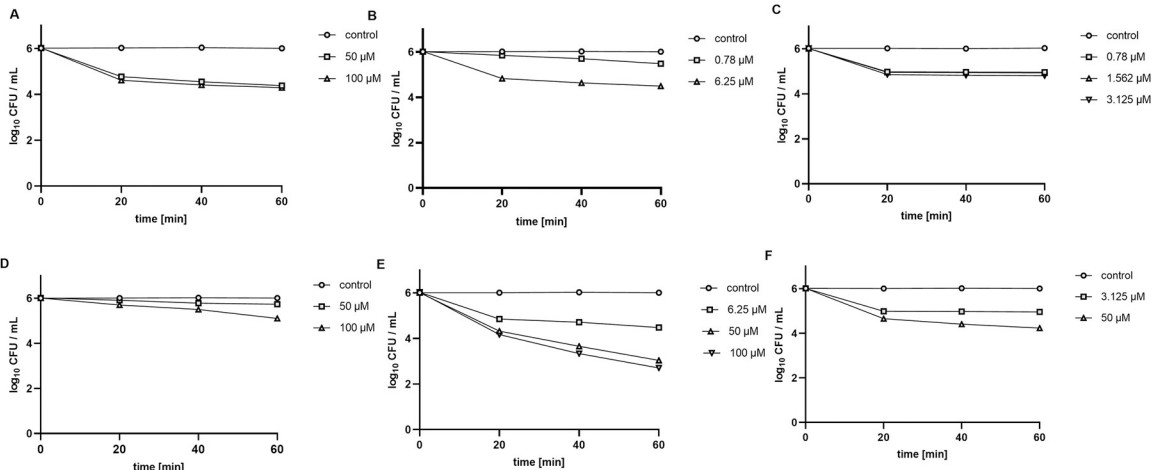

**Fig 6.** 'Dark toxicity' of TMPyP3 on *P. aeruginosa* (A, D), *K. pneumoniae* (B, E) and *K. pneumoniae* OXA-48 (C, F) in tap water (A-C) and municipal wastewater (D-F). Bacterial inactivation with TMPyP3 in different concentrations measured every 20 min over 60 min without any exposure to light. Control is without TMPyP3. Results are shown as an average of multiple measurements ± SD.

(Fig 6D). On both *Klebsiella* strains TMPyP3 showed strong toxicity without irradiation, therefore much lower concentrations were used. At 2 × MEC concentrations (6.25 μM for *K. pneumoniae* and 3.125 μM for *K. pneumoniae* OXA-48) ~1 $\log_{10}$ CFU/mL decrease was observed after 60 minutes of incubation in sterile tap water (Fig 6B and 6C). In wastewater there was a stronger decline with 2 × MEC TMPyP3 concentration, almost ~4 $\log_{10}$ CFU/ml for *K. pneumoniae*, and ~2 log units for *K. pneumoniae* OXA-48 (Fig 6E and 6F).

In tap water, the complete eradication of *P. aeruginosa* was evident with 0.78 μM (0.5 × MEC) already after 15 min of violet-blue irradiation (light dose 18 J/cm$^2$), and with 0.39 μM (0.25 × MEC) after 30 min of irradiation (light dose 36 J/cm$^2$), while with the lowest tested TMPyP3 concentration (0.195 μM), a 4 $\log_{10}$ CFU/mL decrease was observed after 60 min of irradiation (Fig 7A). Higher concentrations, although the same relative to the obtained MEC values, were needed to achieve a similar PDI effect on *P. aeruginosa* in wastewater, and complete photoinactivation was detected with 3.125 μM (0.5 × MEC) after only 15 min, with 1.562 μM (0.25 × MEC) after 30 min, and with 0.78 μM after just over 60 min of violet-blue irradiation (Fig 7D).

The complete eradication of *K. pneumoniae* in tap water was determined after applying TMPyP3 concentration of 1.562 μM (1 × MEC) with irradiation for 15 min, and with 0.78 μM (0.5 × MEC) after irradiation for 60 min (Fig 7B). Again, higher PS concentrations, which corresponded to significantly higher MEC values, were required for the same PDI effect in wastewater, and complete photoinactivation of *K. pneumoniae* was observed with 6.25 μM (4 × MEC) after 15 min of irradiation, with 3.125 (2 × MEC) after 30 min, and with 1.562 μM (1 × MEC) >5 $\log_{10}$ CFU/mL decrease was detected after 60 min of irradiation Fig 7E).

For *K. pneumoniae* OXA-48 in tap water, the complete bacterial inactivation was observed with all tested PS concentrations, from 0.39 μM (0.125 × MEC) after 60 min of violet-blue irradiation, to the 1.562 μM (0.5 × MEC) and 3.125 μM (1 × MEC) that showed complete photoinactivation already after 30 and 15 min respectively (Fig 7C). As already seen on previous two bacterial strains, the complete eradication in wastewater was detected at higher concentrations, and 2 × MEC (6.25 μM) after 15 min of irradiation, 1 × MEC (3.125 μM) after 30 min, and 0.5 × MEC (1.562 μM) after 60 min of irradiation (Fig 7F).

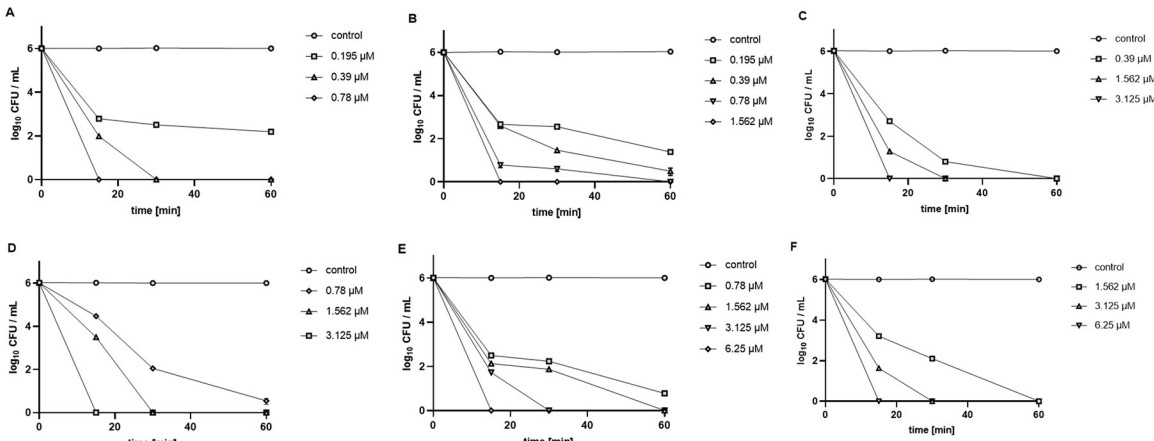

**Fig 7.** Photoinactivation of *P. aeruginosa* (A, D), *K. pneumoniae* (B, E) and *K. pneumoniae* OXA-48 (C, F) with TMPyP3 in tap water (A-C) and municipal wastewater (D-F). Photodynamic inactivation with TMPyP3 in different concentrations and under 0, 15, 30 and 60 min of irradiation with violet-blue light ($\lambda$ = 394 nm, 20 mW/cm$^2$; total light doses 0, 18, 36 and 72 J/cm$^2$ respectively). Control represents samples not treated with TMPyP3. Results are shown as an average of multiple measurements ± SD.

Control                         TMPyP3 treatment

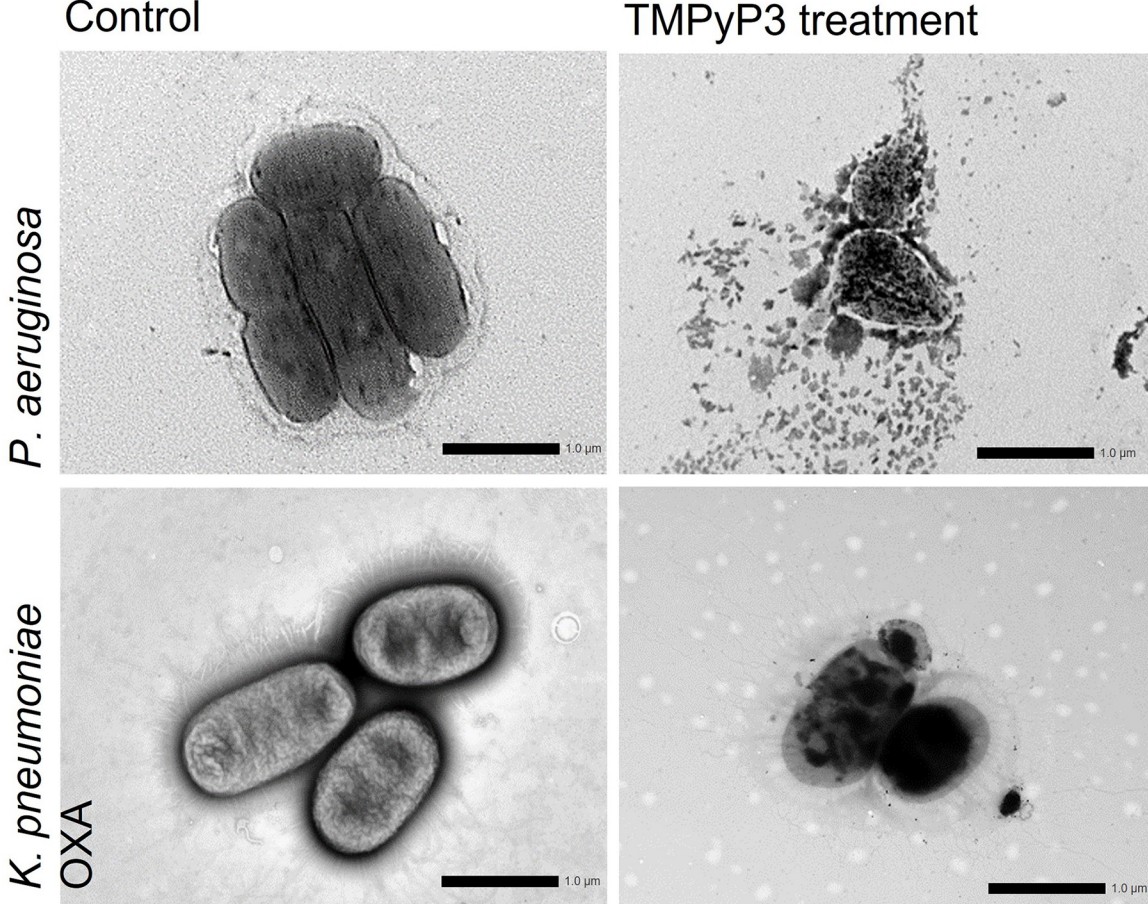

**Fig 8. STEM images of *P. aeruginosa* and *K. pneumoniae* OXA-48 after the treatment with TMPyP3 and violet-blue light.** After 30 min of incubation with TMPyP3 (1 × MEC) bacteria was irradiated with violet-blue light for 10 min (irr. $\lambda$ = 394 nm; 20 mW/cm$^2$;12 J/ cm$^2$). Samples were analysed under STEM after 20 h air-drying at 15k magnification. Control represents non-treated cells.

Finally, the destruction of *P. aeruginosa* and *K. pneumoniae* OXA-48 by PDI was observed by STEM microscopy as the disintegration of the cell wall and leakage of intracellular material (Fig 8).

## Discussion

In our previous spectroscopic analyses of TMPyP3, a small impact of ions present in hard water could be seen on the absorption spectra, and somewhat larger impact was observed on its fluorescence [31]. In this study, in municipal wastewater, a small impact was evidenced by a slight bathochromic shift of the Soret band to 423 nm, compared to 416 nm in DEMI water [31], and 420 nm in tap water [30], accompanied with the lower molar absorption coefficients for all absorption bands. Similarly, lower fluorescence intensity was obtained with TMPyP3 dissolved in wastewater (Fig 2B), compared to hard water, and even lower compared to DEMI water [31]. Furthermore, a moderate stability of TMPyP3 in wastewater was observed (Fig 3 and S1 Dataset), and the measured stability was continuously very similar between the samples, regardless of whether the sample has been illuminated, once or repeatedly, or not at all, suggesting that the PS could be used for multi-day and repeated PDT purposes.

Other authors have already warned of the need to investigate the influence of ions present in the environment where PDI is considered for application, and for example, carbonate and

phosphate ions were reported to decrease the singlet oxygen production of the cationic flavin PS and PDI effect against *S. aureus* and *P. aeruginosa* [37]. On the other hand, it has been shown that although TMPyP4 undergoes complexation with certain cations such as Mg(II), for concentrations of the salts usually present in drinking water, this does not significantly affect $^1O_2$ production [38]. Organic matter present in wastewater (dissolved and suspended) can affect PDI in two, opposite ways; it can enhance it through increased formation of radicals and ROS, or it may reduce it by quenching ROS and/or depleting PS's excited triplet state [10, 18]. For example, dissolved organic matter (DOM), such as fulvic and humic acids, and other humic substances, can generate $^1O_2$ through photosensitisation [32], while nitrite in wastewater and natural organic matter in the water matrix are known to cause oxidant depletion [11]. In our study DPBF was used as a probe that is known to react not only with $^1O_2$, but also with other types of ROS such as hydroxyl radical [39], and a significant photodegradation of DPBF without presence of TMPyP3 (DPBF control) was observed, especially in wastewater (Fig 4). This could be due to the applied irradiation wavelength (411 nm), which is within the absorption spectrum of DPBF, so the dye itself can be a photosensitiser and produce ROS, while the organic matter present in wastewater contributes to this production even more. A significantly higher decrease in fluorescence of DPBF was measured with TMPyP3 in tap water and municipal wastewater, confirming TMPyP3 as a good $^1O_2$ producer, however, relative to the DPBF control, somewhat lower $^1O_2$ production was measured with TMPyP3 in wastewater compared to tap water, suggesting a small negative impact of the present organic matter. Compared to TMPyP4, TMPyP3 also proved to be a better singlet oxygen producer in all three tested water samples (DEMI, tap water and wastewater) (Fig 4, S2 and S3 Datasets), as well as showing lower MEC values on all three bacterial strains studied in wastewater (Table 1B and S3 Table). This is consistent with the results described in the literature, especially those in a recent analysis in which TMPyP3 compared to TMPyP4 proved to be a more stable, better $^1O_2$ producer (by 5.4%), and also significantly more PDI efficient PS on *Escherichia coli in vitro* [29]. It must be noted, that counterion for both porphyrins was 4-methylbenzenesulfonate, while in our study chloride salts were used.

The term violet-blue light (VBL) most commonly refers to a single wavelength of 405 nm [40], but it can include a wavelength range from 395 nm to 420 nm [41]. Closely related, a range from 400 nm to 470 nm is also known as antimicrobial blue light (aBL), and among other applications, it can be used as a standalone disinfectant, against planktonic bacteria and their biofilms [42]. The antibacterial use of VBL and aBL is based on photoactivation of bacterial endogenous porphyrins and, into a lesser extent, other natural PSs such as flavins, thus on their ROS formation and consequent oxidative stress [42]. The main advantages of this method are an higher selectivity and toxicity toward bacterial, compared to human and other cells, and the possibility of applying it in combination with other antimicrobials and disinfectants, as in some cases, in such combinations a synergistic effect has been observed [43], and the absence of the development of selective resistance to this method [44, 45]. In assessing the wavelengths between 375 and 420 nm against biofilms of several nosocomial bacterial strains, the entire range proved effective, but the most efficient wavelengths were those corresponding to porphyrins' Soret band, from 395 nm to 405 nm [41]. Interestingly, several studies showed the strongest response to aBL and VBL 405 in *P. aeruginosa*, while *K. pneumoniae* proved to be the most resistant strain among tested [34, 40], and the observed difference in sensitivity of different bacterial strains was associated with a different composition of endogenous porphyrins [41, 43]. The combination of VBL and indocyanine green (ICG) as exogenous PS, in one previous study was shown to be the most efficient compared to only PDI (with ICG and 810 nm irradiation), or only VBL, against *Streptococcus mutans* and its biofilm, and dual-light simultaneous treatment was particularly effective when daily doses were applied, and with the most

complete effect achieved after 14 days [46]. However, for wastewater-disinfection purposes, VBL wavelengths should be the most efficient part of the solar spectrum, given the lowest water absorption coefficient, and high molar absorption coefficients of many, both exogenous and bacterial endogenous, PSs, in that part of the EM. This is why we chose VBL wavelength (394 nm) to study PDI, with TMPyP3 as an exogenous PS, on MDR strains of *P. aeruginosa* and *K. pneumoniae* in wastewater.

All our inactivation experiments were conducted in tap water for comparison with the results obtained in wastewater, to assess the impact of sanitary waste and organic matter present. In determining the MEC values, the highest photoinactivation index was observed on *P. aeruginosa* in tap water, which may confirm the aforementioned results concerning the greater sensitivity of this strain on VBL compared to *K. pneumoniae*. This index decreased somewhat in wastewater, which can be explained by the poorer survival of *P. aeruginosa* in wastewater, and the ROS depletion by organic matter, but it was achieved again after prolonged illumination. On the other hand, the very low photoinactivation index for both *K. pneumoniae* strains in tap water is consistent with the results from the literature that report a much lower sensitivity to VBL. However, *K. pneumoniae* proved to be very sensitive to added porphyrin (TMPyP3) and for both strains a very low MEC value was measured in tap water without irradiation. A significant difference observed in dark toxicity between *P. aeruginosa* and *K. pneumoniae* may be related to various resistance mechanisms, where *P. aeruginosa* is known to have a strong intrinsic resistance mechanism based on efflux pumps [47]. The dark toxicity of TMPyP3 was significantly lower in wastewater, but after irradiation with VBL, for both strains of *K. pneumoniae*, a much higher photoinactivation index was achieved than in tap water. This was higher for *K. pneumoniae* than *K. pneumoniae* OXA-48 after 10 min of irradiation, however, further irradiation and a higher light dose did not have effect on *K. pneumoniae*, while for *K. pneumoniae* OXA-48 the MEC value obtained after 30 min irradiation was lower and equalized with the MEC value for *K. pneumoniae*. Altogether, low micromolar MEC values for TMPyP3 were obtained for all three bacterial strains after short illumination (10 min) and with very similar values between tap water and wastewater, or, in wastewater, after prolonged illumination (30 min). Significantly higher differences of dark toxicity of TMPyP3 between tap water and wastewater might even be promising for the application because they indicate that the present waste substances can reduce the PS's dark toxicity, without notably affecting its PDI effect. Moreover, a slight decrease of PDI in wastewater can be compensated by slightly longer illumination. These results also confirm the importance of conducting measurements with as similar conditions as expected in application. All the MAAC values determined after 10 min of irradiation are very close to the respective MEC values for each strain, or even slightly lower, which speaks of the potential of this method also in biofilm control. Numerous results from the literature indicate that PDI could be more effective in treating wastewater if lower concentrations of the PS were used with low light fluence over a longer irradiation period [22–24]. Our PDI kinetics experiments have shown that the complete eradication could be achieved for all three bacterial strains already with 1.562 μM of TMPyP3 and 60 min of irradiation (72 J/cm$^2$) with VBL.

Two members of the ESKAPE group, *K. pneumoniae* and *P. aeruginosa* are both Gram–negative, rod-shaped bacteria that are ubiquitously found in different environments such as soil, water, vegetation and material formed by organic decay, both are known for their ability to adapt and survive even in the extreme environment, and also as opportunistic pathogens of plants, animals and humans, and responsible for high number of HAI [6, 48]. *K. pneumoniae* is an atmospheric nitrogen-fixing bacterium, it has a complex catabolism and a metabolic diversity, and even MDR clones share many of these metabolic features [48, 49], with a strong ability to thrive in its environment [50], which could also be seen in our study in survival tests.

There are already several reports of MDR strains reaching environment, and for example, ESBL-resistant *K. pneumoniae* was found surviving in hospital wastewater even after the standard wastewater treatment [51]. According to the literature, there seems to be few studies dealing with the photoinactivation of *K. pneumoniae*, and of these, the most used PS is methylene blue, or 5-aminolevulinic acid (ALA) in high concentrations and doses of white light [52]. To the best of our knowledge, our study is the first evaluation of PDI on *K. pneumoniae* in wastewater, and by using an exogenous porphyrin. Surprisingly, the high dark toxicity of TMPyP3 on *K. pneumoniae* was observed in tap water, but in the municipal wastewater sample, the sensitivity of both *K. pneumoniae* strains to TMPyP3 without light activations was much lower. It must be noted that during all our experiments, in which we tested toxicity in the dark, great care was taken to maximally protect them from light. However, it has been shown with strong $^1O_2$ producers, e.g. with TMPyP4, that even very short exposure to ambient, even very low light, may be partially responsible for the observed 'dark toxicity', which has been reported for several bacteria [53]. On the other hand, very low toxicity without light was shown with TMPyP3 on *P. aeruginosa*, and a strong PDI effect upon combination with VBL. Previously, TMPyP3 was tested against *P. aeruginosa* with the PS immobilised on cellulosic fabric [54], TMPyP4 was used in combination with white light against clinical isolates of planktonic *P. aeruginosa* and in the biofilm [55, 56], and only cationic diaryl-porphyrins were combined with VBL and proved promising against the *P. aeruginosa* PAO1 biofilm [33]. In our study on *P. aeruginosa* a high photoinactivation index was demonstrated using TMPyP3 in combination with VBL in tap water, which can also be achieved in municipal wastewater, if necessary, with slightly longer illumination depending on the amount of organic and inorganic matter present. For both *P. aeruginosa* and *K. pneumoniae* OXA-48, STEM images showed a complete destruction of the bacterial cell due to PDI with TMPyP3, and in the case of *P. aeruginosa* more pronounced damage to the cell wall and consequent leakage of intracellular material (Fig 8). While for effective antitumor PDT, the entry of PS into the target cell is important, for bacterial cells this is not necessary, but rather strong enough binding on the surface [57]. It is known that cationic PSs bind particularly well to the surface of Gram-negative bacteria thanks to electrostatic interactions, which was confirmed for TMPyP3 in our previous work when it was shown that it binds equally well to the surface of *L. pneumophila*, already after 10 minutes, as more lipophilic TMPyP3-C$_{17}$H$_{35}$ [30].

## Conclusions

The aim of this study was, on the one hand, to investigate the influence of wastewater constituents on the properties of photosensitizer (TMPyP3) and its PDI effect, and, on the other hand, to compare the sensitivity of different MDR Gram-negative bacterial strains, for which new solutions in wastewater treatment are sought, to PDI with an exogenous PS. Therefore, *P. aeruginosa*, a bacterium sensitive to violet-blue light alone, was chosen for comparison with *K. pneumoniae*, which is significantly less sensitive to VBL, and for which the effect of exogenous porphyrin and its PDI in wastewater has not been described so far. *P. aeruginosa* and *K. pneumoniae* showed significantly different sensitivity to TMPyP3 without irradiation, however, a significant, and similar PDI effect was achieved on all three bacterial strains after irradiation with VBL. Wastewater constituents showed only a small negative impact on PDI, probably due to ROS depletion and light attenuation, but were also responsible for lower toxicity of TMPyP3 on both *K. pneumoniae* strains, which can be considered a positive impact in terms of PDI. Photoinactivation of MDR bacteria by cationic porphyrins such as TMPyP3, therefore, seems a very promising method in combination with VBL because it can target bacteria of different susceptibility to exogenous porphyrins, as well as different sensitivities to VBL.

Municipal wastewater could be treated with an adjusted concentration of the photosensitiser through an appropriate time of (repeated) irradiation, depending on the bacteria and waste substances present. Finally, VBL is also part of the solar spectrum, so sunlight could be used for photoactivation and sustainable application in wastewater disinfection.

## Supporting information

**S1 Table. Physicochemical parameters of municipal wastewater sample collected and analysed at the Teaching Institute of Public Health of Primorje-Gorski Kotar county.**
(PDF)

**S2 Table. Antibiotic susceptibility testing.**
(PDF)

**S3 Table. Minimal effective concentration (MEC) values of TMPyP4 on *P. aeruginosa*, *K. pneumoniae* and *K. pneumoniae* OXA-48 in municipal wastewater without light (dark) and with irradiation ($\lambda$ = 394 nm; 20 mW/cm$^2$) for 10 min.**
(PDF)

**S1 Dataset. Photostability of TMPyP3 in municipal wastewater measured in a 5-day experiment.**
(XLSX)

**S2 Dataset. Kinetics of singlet oxygen production in DEMI water and municipal wastewater.**
(XLSX)

**S3 Dataset. Singlet oxygen production in tap water.**
(XLSX)

**S4 Dataset. Tabular results of survival, dark toxicity and photoinactivation by TMPyP3 of *P. aeruginosa*, *K. pneumoniae* and *K. pneumoniae* OXA-48.**
(XLSX)

## Acknowledgments

We would like to thank Dr Ivna Kavre Piltaver from the Faculty of Physics and Centre for Micro- and Nanosciences and Technologies at the University of Rijeka for the help with the STEM analyses, and the members of the Workshop of Physical Chemistry Division at the Ruder Bošković Institute in Zagreb for the light sources used in this work.

## Author Contributions

**Conceptualization:** Nela Malatesti, Ivana Gobin.

**Formal analysis:** Martina Mušković, Matej Planinić, Antonela Crepulja, Marko Lušić, Marin Glad.

**Funding acquisition:** Nela Malatesti, Ivana Gobin.

**Investigation:** Martina Mušković, Matej Planinić, Antonela Crepulja, Marko Lušić.

**Methodology:** Martina Mušković, Marin Glad, Martin Lončarić, Nela Malatesti, Ivana Gobin.

**Project administration:** Nela Malatesti, Ivana Gobin.

**Resources:** Marin Glad, Martin Lončarić, Nela Malatesti, Ivana Gobin.

**Supervision:** Nela Malatesti, Ivana Gobin.

**Validation:** Martina Mušković, Matej Planinić, Antonela Crepulja, Marko Lušić, Marin Glad, Martin Lončarić, Nela Malatesti, Ivana Gobin.

**Visualization:** Martina Mušković.

**Writing – original draft:** Martina Mušković, Nela Malatesti, Ivana Gobin.

**Writing – review & editing:** Martina Mušković, Matej Planinić, Antonela Crepulja, Marko Lušić, Marin Glad, Martin Lončarić, Nela Malatesti, Ivana Gobin.

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
