## [Decision Letter · Decision Letter 0]

6 Jun 2023

PONE-D-23-13741Photoinactivation of multidrug-resistant strains of Klebsiella pneumoniae and Pseudomonas aeruginosa in municipal wastewater by tetracationic porphyrin and violet-blue lightPLOS ONE

Dear Dr. Malatesti,

Thank you for submitting your manuscript to PLOS ONE. After careful consideration, we feel that it has merit but does not fully meet PLOS ONE’s publication criteria as it currently stands. Therefore, we invite you to submit a revised version of the manuscript that addresses the points raised during the review process.  - The novelty is not emphasized here.

- The main reason for the use of a violet-blue light source Why not sunlight.

 - The production of singlet oxygen should be compared with another photosensitizer as a reference in the same medium and the obtained results of inactivation should also be compared with another photosensitizers.

-  Describe the statistical significance of the values in figures.

- A better discussion about the importance of organic and inorganic matter in the photodynamic efficiency.

- The discussion should be rewritten, it is hard to follow.

. The conclusion must also be improved since does not reflect what was attained.

We look forward to receiving your revised manuscript.

Kind regards,

Adelaide Almeida

Academic Editor

PLOS ONE

Journal Requirements:

3. Please expand the acronym “MZO” (as indicated in your financial disclosure) so that it states the name of your funders in full.

Additional Editor Comments:

- The novelty is not emphasized here.

- The main reason for the use of a violet-blue light source Why not sunlight.

- The production of singlet oxygen should be compared with another photosensitizer as a reference in the same medium and the obtained results of inactivation should also be compared with another photosensitizers.

- Describe the statistical significance of the values in figures.

- A better discussion about the importance of organic and inorganic matter in the photodynamic efficiency.

- The discussion should be rewritten, it is hard to follow.

. The conclusion must also be improved since does not reflect what was attained.

Reviewers' comments:

Reviewer's Responses to Questions

**Comments to the Author**

1. Is the manuscript technically sound, and do the data support the conclusions?

Reviewer #1: Partly

Reviewer #2: Yes

Reviewer #3: Yes

Reviewer #4: Yes

Reviewer #5: Partly

2. Has the statistical analysis been performed appropriately and rigorously? 

Reviewer #1: No

Reviewer #2: I Don't Know

Reviewer #3: Yes

Reviewer #4: Yes

Reviewer #5: No

3. Have the authors made all data underlying the findings in their manuscript fully available?

Reviewer #1: Yes

Reviewer #2: Yes

Reviewer #3: Yes

Reviewer #4: Yes

Reviewer #5: No

4. Is the manuscript presented in an intelligible fashion and written in standard English?

Reviewer #1: No

Reviewer #2: Yes

Reviewer #3: Yes

Reviewer #4: Yes

Reviewer #5: No

5. Review Comments to the Author

Reviewer #1: The article entitled “Photoinactivation of multidrug-resistant strains of Klebsiella pneumoniae and Pseudomonas aeruginosa in municipal wastewater by tetracationic porphyrin and violet-blue light” by Nela Malatest reports the effect of a tetracationic symmetric porphyrin (TMPyP3) without light, and with activation by violet-blue light (VBL), on MDR strains of Pseudomonas aeruginosa, Klebsiella pneumoniae and K. pneumoniae OXA-48 in tap water and municipal wastewater.

Although the study is interesting the article require some clarifications and improvments.

Concerning the English although in general is fine sometimes is difficult to understand the message of determined comments.

i) The title must be revised in order to reflect what was studied.

ii) Why the authors selected (TMPyP3) with violet-blue light to perform these studies? What was the motivation? Although this is implicit in the discussion this aspect must appear earlier namely at the end of the introduction

iii) In the abstract the full name of TMPyP3 must be indicated and also of other abbreviations (e.g. VDL)

iv) Check the value of the concentration of porphyrin (1,562 uM)?

v) Rephrase the following comment since is not clear “Although first discovered on a microorganism, PDT was developed primarily as antitumor therapy (1).”

vi) What the authors want to say in the following comment “One of the main reasons for seeking new antimicrobial approaches in aPDT is that excessive and ……“ with for seeking new antimicrobial approaches in aPDT

vii) Is not clear at the end of the introduction what the authors are going to do. First they refer that are going to evaluate the efficiency of TMPyP3 in waste water and then in tap water. The authors must try to be more specific showing clear what they want to evaluate and why.

viii) The RMN of the porphyrin must be checked and the numbers of the protons added in the figure. The authors must be aware that protons that are coupling with each other must have the same coupling constant. So all the coupling constants must be revised and corrected.

ix) Line 103 Concerning the following comment “A stock solution of TMPyP3 for microbiological experiments was prepared in sterile tap water (200 µM).” what the authors want to mean with sterile tap water?

x) Line 110 However in this section the authors refer that Stock solution of TMPyP3 was prepared in demineralised (DEMI) water and diluted in (waste)water samples until the final concentration .. . All this seems a bit confuse. Why not always in the same conditions?

xi) In line 209 the authors refer that The photophysical and photochemical properties of TMPyP3 in the wastewater sample were evaluated by absorption and fluorescence spectroscopy, and (photo)stability, and singlet oxygen production (1O2) were measured to examine the influence of the inorganic and organic matter contained in water. However, in the experimental part the authors refer that before the experiments, wastewater was filtered and sterilised by autoclaving (121°C, 1.2 bar, 147 20 min). Please clarify this aspect and indicate the type of filter used.

xii) In fig 4 is not clear in the Y axis that what is being observed is a decrease in fluorescence

xiii) In the discussion concerning the achievments already obtained with other porphyrins it would be more easy to follow if the authors introduce their structures in the same figure of TMPyP3.

xiv) During the discussion the authors must refer the previous figures presented in the results and related with the subject that they are commenting in order to facilitate their comprehension.

xv) Line 402 The following comment is a bit confuse and must be clarified “should be noted that high photodegradation of DPBF without presence of TMPyP3 was also observed, especially in wastewater, which could be due to overlap of the applied irradiation wavelength (411 nm) with the absorption spectra of DPBF, and the resulting photochemical reaction that is enhanced in the presence of organic matter”

xvi) No errors can be visualized in graphics of figure 7

xviii) Line 466 The following comment is difficult to accept :Surprisingly, the high dark toxicity of TMPyP3 on K. pneumoniae was observed in tap water, but in the municipal wastewater sample, the sensitivity of both K. pneumoniae strains to TMPyP3 without light activations was much lower. It must be noted that during all our experiments it was not possible to exclude absolutely all light, and it was shown, e.g., with TMPyP4, that even ambient light during the experiment can be responsible for the observed ‘dark toxicity’, which was reported for several bacteria (47).” So the authors are trying to say that it is difficult to exclude all light so that toxicity is due to the photodynamic action of porphyrins? Usually the systems can be easily protected from light by covering with aluminium foil. Concerning the comment with TMPyP4, that even ambient light during the experiment can be responsible for the observed ‘dark toxicity’, which was reported for several bacteria what the authors want to say? If is at ambient light they can not say that it is observed dark toxicity.

xix) The discussion is really hard to follow and to facilitate the reading the authors must introduce Figures 8 -10 in this section. It is not clear the real achievements obtained in these studies. So this section requires a real improvement.

xx) The conclusion must also be improved since does not reflect what was done and what was attained.

Reviewer #2: General comments

In this study, Muskovic et al evaluated the effects of Antimicrobial Photodynamic Treatment (APDT) with a tetracationic porphyrin and violet-blue light on Klebsiella pneumonia and Pseudomonas aeruginosa bacteria, both in tap and wastewater. The results showed that APDT is able to inactivate planktonic cells of both species in both situations. The work is well written, presented and the results are discussed appropriately. However, the effects of APDT with various photosensitizers, including porphyrins, on these bacterial species, both in tap and wastewater, have already been described previously. Thus, the main weakness of the work is that the authors do not make it clear, from the beginning of the manuscript, what is the novelty of their studies and how they contribute to the advancement of APDT.

Specific comments

Abstract

Lines 15-17. Rewrite.

Introduction

Lines 148-49. Rewrite.

Materials and Methods

Lines 130-133. Rewrite.

Line 146. How was the filtration performed?

Lines 158, 165 and 180. How many biological replicates were performed in each of the experiments? This needs to be stated clearly.

Results.

Line 299. Change to “continued to increase”

Discussion

The discussion is too long and could be shortened.

Lines 447-449. This is predominantly results (including Fig. 11) and should be moved to the “Results” section

Conclusions

Lines 478-479. The first sentence is not a conclusion and should be removed.

Reviewer #3: Review of „Photoinactivation of multidrug-resistant strains of Klebsiella pneumoniae and Pseudomonas aeruginosa in municipal wastewater by tetracationic porphyrin and violet-blue light.” By Martina Musckovic et al.

The paper describes the effect of cationic porphyrin (TMPyP3) activated with violet-blue light (VBL) on three strains of Gram-negative bacteria. The experiments were properly performed using standard techniques, and the conclusions are sound and correct. However, the novelty is not emphasized here.

The authors performed photoinactivation experiments on liquid bacterial cultures dispersed in tap water and compared them to cells dissolved in wastewater, finding that all three strains could be photoinactivated with relatively low PS concentrations and low light doses. Although I believe that PDI could be used for wastewater treatment to prevent the spread of MDR bacteria, I feel it is a limitation of this manuscript to work with only one compound. The experiments performed answer the question of whether TMPyP3-PDI kills bacteria but say nothing about the mechanism of bacterial death. Figure 11 would be a good starting point to explain the mechanism of bacterial death, but more is not known or discussed.

I would expect the manuscript to be greatly shortened, especially the discussion section, which is largely a repetition of the results. Instead, the manuscript would benefit greatly if the authors compared the efficacy of the approach studied with others based on different PS. This could be done not necessarily based on experiments conducted by the authors but by other groups as well. Such a comparison would give the reader an idea of which properties of PS are beneficial in treating wastewater. Furthermore, it is not clear to me why the authors chose this particular PS for the studies.

Specific comments

The abstract is too general, I would encourage the authors to be more specific, write a number instead of the term "high", state the concentration instead of writing "at the same concentration". It is not entirely clear what is meant here by "controlled fluences and light doses" (p. 2 lane 27-29). Is it not the case that scientists always use precise doses of light for a particular application?

Authors use several expressions for photodynamic treatment: PDI, aPDT, PDT – that could be confusing.

It is not clear whether the authors use MDR to describe the phenomenon of multidrug resistance or strains that are multidrug resistant.

‘ESKAPE bacteria and their biofilms’ – did you mean: bacteria and biofilm they produce?

Authors also use terms: pre-treatment, primary and secondary treatment in relation to wastewater, but this is not explained at all and is unknown to non-specialist readers.

I would not fully agree with the statement that protein inactivation leads to ARG destruction (p. 3, pp. 67-69). Such a mental shortcut could be misleading. Amino acids and proteins are the first biomolecules exposed to ROS, probably because of their abundance in cells, and DNA and RNA could also be destroyed if PS comes close, however, destruction of ARGs is a side effect of DNA damage when it occurs but is not targeted.

“Water applications" - unclear (p. 4 lane84).

“Enterococcus" - did you mean Enterococcus spp. or the entire genus?

Please provide an antibiotic resistance profile for the strains studied.

Figures 8, 9 and 10 should be combined.

Reviewer #4: In this study the authors report photodynamic effect of tetracationic water soluble porphyrin, 5,10,15,20-(pyridinium-3-yl) porphyrin tetrachloride (TMPyP3) using violet-blue light as light source, against of P. aeruginosa and K. pneumoniae in municipal wastewater. The aim of the work is highly ambitious, and the results are well explored and are, indeed, very promising. The use of aPDT in wastewater disinfection is the subject of several research groups, but has encountered some obstacles for its implementation in the field. These hindrances were mentioned by the authors, however, there are many points that need to be better discussed:

- It is not clear the main reason for the use of a violet-blue light source, mainly its advantage in practical field. It would not be more cost-effective the use of sunlight or even a white light source?

- Authors referred organic and inorganic matter could be important factors in the photodynamic efficiency, however, no discuss are made to clarify this point. How can those chemicals interferer with photodynamic efficiency? The organic and inorganic municipal wastewater herein used was analyzed?

- It is not clear whether TMPy3 could be removed from the water after treatment. In a real application scenario, it is possible the use of a photostable and non-immobilized porphyrin as antimicrobial agent that will be released into natural waters? Do the authors have any data that could elucidate this point?

Reviewer #5: General comments

This manuscript describes some properties of a tetracationic porphyrin named TMPyP3 and the photoinactivation of Klebsiella pneumoniae and Pseudomonas aeruginosa in municipal wastewater irradiated with violet-blue light. The porphyrin used is a conventional photosensitizer studied for more than 25 years (eg see JPPB, 1996, 35, 149) in the PDI of microorganisms and other applications. In its present form, the manuscript is not uninteresting and does not add much to previous knowledge on the topic developed. This is also manifested in the ambiguous conclusions of the work.

Specific comments

- Abstract. The first introductory sentences should be considerably shortened,

- Abstract. “tetracationic symmetric porphyrin” what does symmetric mean? It is unclear.

- Abstract. What does "(1,562 uM)” mean?

- Introduction. Several significant contributions in PDI with tetracationic porphyrins are missing.

- Introductio. “light, which also penetrates best through water.” Water does not absorb visible light. It should be explained.

- The name of TMPyP3 should be correctly given the first time it is named and then use its abbreviation.

Why was TMPyP3 selected as a photosensitizer? The studies should be compared with those obtained for active photosensitizers under the same experimental conditions.

- How was exactly the stock solution of TMPyP3 prepared?

- Absortion and fluorescence emission spectra should be compared with those previously reported in different media.

- In singlet oxygen determinations, the kinetic results should be given (60 s for 10 minutes.). Also, the spectral changes of DPFB should be shown in supporting information.

- The production of singlet oxygen should be compared with another photosensitizer as a reference in the same medium.

- Figure 4 should be better described and explained in the text.

- Table 1 and Table 2. What is the actual number of significant figures in the values?

- Figures 5-7. What is the statistical significance of the values? What is the error of each determination?

- If the purpose is to decontaminate wastewater from microorganisms, why was white light of natural origin not used?

- How were the biofilms prepared?

- The discussion is extremely poor and lacks comparison with previous studies.

- Conclusions are unclear and imprecise.

6. PLOS authors have the option to publish the peer review history of their article (what does this mean?). If published, this will include your full peer review and any attached files.

Reviewer #1: No

Reviewer #2: **Yes: **Gilberto Úbida Leite Braga

Reviewer #3: No

Reviewer #4: No

Reviewer #5: No

---

## [Author Response · Author response to Decision Letter 0]

10 Jul 2023

Dear Editor, Professor Almeida,

Thank you for giving us the opportunity to revise our manuscript “Photoinactivation of multidrug-resistant strains of Klebsiella pneumoniae and Pseudomonas aeruginosa in municipal wastewater by tetracationic porphyrin and violet-blue light”, submitted to PLOS ONE (PONE-D-23-13741).

The manuscript is now revised according to the journal requirements, and the name of the authority that approved the field site access is added in the Methods and materials section, as well as the expanded acronym for MZO (the Ministry of Science and Education of Croatia, or in Croatian: Ministarstvo znanosti i obrazovanja) under Funding.

Please find our responses to yours and the reviewers’ comments:

Editor Comments: 

- The novelty is not emphasized here. 

RE: Among ESKAPE bacteria, Pseudomonas aeruginosa and Klebsiella pneumoniae are both Gram-negative, opportunistic pathogens, which are high on the global priority list, set by WHO, of antibiotic-resistant bacteria that require innovative approaches. However, unlike P. aeruginosa, K. pneumoniae is still underrepresented in the literature in the context of photodynamic inactivation (PDI). Most of these works deal with the treatment of infections caused by this bacterium, mostly with the use of 5-aminolevulinic acid and its derivatives. To our knowledge, so far PDI has not been described using exogenous porphyrin on K. pneumoniae in wastewater, and in the revised paper we tried to emphasize this better. Our PDI results with 5,10,15,20-(pyridinium-3-yl)porphyrin (TMPyP3) were obtained on two multidrug-resistant (MDR) strains of K. pneumonia in tap water and wastewater, and they show that PDI is a very promising method for their treatment in wastewater disinfection, and worth further research. Furthermore, we used MDR P. aeruginosa for comparison with K. pneumonia because of their different sensitivities to violet-blue light (VBL) that we used in our study for photoinactivation of TMPyP3, and through PDI kinetics studies, we found conditions in which all three bacterial strains can be effectively destroyed by photoactivated TMPyP3.

Impact of wastewater constituents is clearly very important for the overall PDI effect; however, we were surprised to find recently in a review that of the nearly 11,000 publications describing the inactivation of bacteria by different photo-disinfection processes in water, only 4.4% of them described water matrix and reported the effect of water constituents (Gandhi et al. 2023). In our study we evaluated the impact of wastewater constituents on TMPyP3 and its stability, singlet oxygen production, and PDI. Given that in our previous work we analysed the impact of inorganic ions present in tap water on TMPyP3 and its PDI effect, and since tap water is part of the municipal wastewater, in this work we used tap water for comparison to emphasize the impact of organic matter present in wastewater. While turbidity is known to have negative effect due to light attenuation, other water constituents, such as dissolved organic matter (DOM), can have both negative and positive effects. In our work we have confirmed small negative effect in terms of somewhat higher MEC values obtained in wastewater. However, phototoxic index (the difference between PS’s activity with light irradiation and without light), proved to be better in wastewater for both K. pneumoniae strains.

We also noticed that in studies like this, bacterial survival in the tested medium is very rarely described, which we consider important to check before photoinactivation studies. We conducted a survival study with all three bacteria over 14 days in tap water as well as in wastewater, and found that all three bacterial strains easily survive in these conditions throughout the period, which confirms that it is necessary to look for methods to combat these pathogens in such environments.

- The main reason for the use of a violet-blue light source Why not sunlight.

RE: We are aware that for the application of photoinactivation in wastewater disinfection, it would be best to use sunlight for photoactivation in terms of sustainability. However, although numerous examples of antimicrobial PDT can be found in the literature, and this number is constantly growing, it is increasingly pointed out that the literature results of PDI are difficult to compare because different light sources, fluence rates and doses of light, together with different concentrations of photosensitisers (PSs) and different pathogens are used (Phasupan et al. 2021, Piksa et al 2023). To compare the effect of PSs on different bacteria, it is useful to know the absorbed photons, which can be calculated if we know the spectral characteristics of both PS and light sources (Phasupan et al. 2021, Aroso et al. 2021), and it is important to have as precise optical and biological parameters as possible in in vitro studies (Piksa et al. 2023) to better optimize PDI for different applications later and on different pathogens. In solar irradiation experiments, the average fluence rate can be obtained, but since we wanted to focus on the influence of wastewater constituents, and considering that the composition of wastewater is complex and can vary, in this study we wanted to have all the other parameters specified and as simple as possible to monitor. Therefore, we decided to monitor the kinetics of photoinactivation with different concentrations of TMPyP3 using the same fluence rate of a certain wavelength. We decided on a wavelength of 394 nm for several reasons, which we stated in the manuscript, and in the revised paper we tried to emphasize and clarify this even more. Violet-blue light (VBL) is part of the solar spectrum, so its impact can be expected in applications with the use of solar irradiation, and these are also wavelengths that penetrate the deepest through the water. In fact, water absorption spectrum shows the lowest absorption coefficient at VBL wavelengths, and for example, red light absorbs 100 times more (S. Prahl, Optical absorption of water. Available at http://omlc.ogi.edu/spectra/water/index.html; accessed 03 July 2023). Many exogenous PSs have good absorption at VBL wavelength, and we used 394 nm also for TMPyP3 in our previous work where we examined the influence of ions present in tap water on PDI. Furthermore, VBL is the most effective part of the solar spectrum for photoinactivation of bacterial endogenous PSs (Halstead et al. 2016). Therefore, one of the aims in this study was also to compare the susceptibility of different Gram-negative strains from the group of ESKAPE bacteria to PDI with exogenous porphyrin by using VBL. One of the problems of existing water disinfectant methods is the different sensitivity of individual strains, which may lead to even greater resistance of those that are much less susceptible. For this reason, we chose Pseudomonas aeruginosa for comparison, for which a number of examples of PDI can be found in the literature, and which is known to be sensitive to VBL alone (i.e. without the presence of exogenous PS). On the other hand, we chose Klebsiella pneumoniae, which is known to be significantly less sensitive to VBL, and for which there is much less literature data in the context of PDI, mostly for treating infections, and to our knowledge, has not been described so far by the use of exogenous porphyrin in wastewater.

- The production of singlet oxygen should be compared with another photosensitizer as a reference in the same medium and the obtained results of inactivation should also be compared with another photosensitizers.

RE: We have again conducted experiments to measure singlet oxygen production using 5,10,15,20-(pyridinium-4-yl)porphyrin tetrachloride (TMPyP4), a very well-known and well-described photosensitiser, as a reference and for comparison with TMPyP3. The results obtained for DPBF control, TMPyP3 and TMPy4 in DEMI water and wastewater are in Fig 4, and all the kinetics results from that experiment are in the Supporting information (S2 Dataset). We also added the results for new experiment performed with these compounds in tap water, and they are in the Supporting information (S3 Dataset). In all these conditions, TMPyP3 showed somewhat better singlet oxygen production (Fig 4), which agrees with some other literature results that we cited in our paper. In addition, we determined the MEC values for TMPyP4 on all three bacterial strains in wastewater, under the same conditions as for TMPyP3 (with light irradiation for 10 min, and without light), and the obtained MEC values are also somewhat higher for TMPyP4 (S3 Table), in accordance with 1O2 production. Comparisons of our PDI results with the most relevant results from the literature are also emphasized in the discussion.

- Describe the statistical significance of the values in figures. 

RE: In the Materials and methods section, information on the number of repetitions of microbiological experiments is added to all relevant subsections. Moreover, S4 Dataset with all survival, dark toxicity and photoinactivation results for all three bacteria (shown in Figs 5, 6 and 7), with all calculated average and standard deviation values is added in the Supporting information.

- A better discussion about the importance of organic and inorganic matter in the photodynamic efficiency.

RE: One of the main aims of our study was to investigate the influence of wastewater constituents on the properties of photosensitizer and PDI effect. Given that in our previous two papers (which we refer to and cite in the discussion) we analysed the impact of inorganic ions present in tap water on TMPyP3 and its PDI effect, and since tap water is part of the wastewater, in this work we used tap water for comparison to emphasize the impact of organic matter present in wastewater. We added this clarification and revised that part accordingly in the introduction. In addition to the already conducted analysis of the physicochemical properties of wastewater, we have added the results of determining the physicochemical properties of wastewater after filtration and sterilization (S1 Table). In accordance with these results, we revised the discussion and tried to better emphasize the observed impact of wastewater constituents on PDI effect.

- The discussion should be rewritten, it is hard to follow. 

RE: The discussion is shortened and thoroughly revised, in line with comments and suggestions from all reviewers.

- The conclusion must also be improved since does not reflect what was attained. 

RE: The conclusion was revised to be clear and more precise, and to emphasize which studies were conducted, why, and what their main findings were.

Reviewer #1: The article entitled “Photoinactivation of multidrug-resistant strains of Klebsiella pneumoniae and Pseudomonas aeruginosa in municipal wastewater by tetracationic porphyrin and violet-blue light” by Nela Malatesti reports the effect of a tetracationic symmetric porphyrin (TMPyP3) without light, and with activation by violet-blue light (VBL), on MDR strains of Pseudomonas aeruginosa, Klebsiella pneumoniae and K. pneumoniae OXA-48 in tap water and municipal wastewater. Although the study is interesting the article require some clarifications and improvements. Concerning the English although in general is fine sometimes is difficult to understand the message of determined comments.

RE: We are thankful to the Reviewer for all the comments and suggestions according to which we have made changes to the manuscript as follows.

i) The title must be revised in order to reflect what was studied.

RE: We propose an extension of the title that emphasizes the aim of the study, so the revised title would be: “Photodynamic inactivation of multidrug-resistant strains of Klebsiella pneumoniae and Pseudomonas aeruginosa in municipal wastewater by tetracationic porphyrin and violet-blue light: the impact of wastewater constituents”

ii) Why the authors selected (TMPyP3) with violet-blue light to perform these studies? What was the motivation? Although this is implicit in the discussion this aspect must appear earlier namely at the end of the introduction

RE: We explained in detail in the discussion why we chose TMPyP3 for this study, but we agree with the Reviewer that it should be even clearer and stated much earlier in the text, so according to the Reviewer’s suggestion we moved this explanation to the last part of the introduction.

iii) In the abstract the full name of TMPyP3 must be indicated and also of other abbreviations (e.g. VDL)

RE: The full name of TMPyP3 is given (5,10,15,20-(pyridinium-3-yl)porphyrin tetrachloride) the first time it is named in the abstract. All other abbreviations are used after the first indication of the full name.

iv) Check the value of the concentration of porphyrin (1,562 uM)?

RE: This has now been corrected with the appropriate unit (�M).

v) Rephrase the following comment since is not clear “Although first discovered on a microorganism, PDT was developed primarily as antitumor therapy (1).”

vi) What the authors want to say in the following comment “One of the main reasons for seeking new antimicrobial approaches in aPDT is that excessive and ……“ with for seeking new antimicrobial approaches in aPDT

RE: Both sentences are now rephrased.

vii) Is not clear at the end of the introduction what the authors are going to do. First they refer that are going to evaluate the efficiency of TMPyP3 in waste water and then in tap water. The authors must try to be more specific showing clear what they want to evaluate and why.

RE: One of the main aims of our study was to investigate the influence of wastewater constituents on the properties of photosensitizer and PDI effect. Given that in our previous two papers (which we refer to and cite in the discussion) we analysed the impact of inorganic ions present in tap water on TMPyP3 and PDI, and since tap water is part of the wastewater, we used tap water for comparison to emphasize the impact of organic matter present in wastewater. We added this clarification and revised that part accordingly in the introduction.

viii) The RMN of the porphyrin must be checked and the numbers of the protons added in the figure. The authors must be aware that protons that are coupling with each other must have the same coupling constant. So all the coupling constants must be revised and corrected.

RE: The reviewer is right and we are grateful to the reviewer for pointing out this mistake. The signal for protons at the 5-position of pyridyl units (Py-5-H) appears to be a triplet instead of an expected doublet of doublets due to the overlap of sub-peaks, so it is not possible to obtain the correct coupling constants. Accordingly, we have corrected the signal assignment. We also checked the numbers of all protons and they are correct.

ix) Line 103 Concerning the following comment “A stock solution of TMPyP3 for microbiological experiments was prepared in sterile tap water (200 µM).” what the authors want to mean with sterile tap water?

RE: Sterile tap water was prepared by autoclaving (121°C, 1.2 bar, 20 min) the tap water taken in our laboratory. After dissolving the porphyrin in water, the solution was also filtered. We added information about autoclaving and about the filter used, so now it reads: “A stock solution of TMPyP3 (200 µM) for microbiological experiments was prepared in sterile tap water (sterilised by autoclaving at 121°C, 1.2 bar, 20 min). After dissolving porphyrin in water, the solution was filtered using sterile Syringe Filter 0,45 �m (Labex Ltd, Budapest, Hungary), and kept covered in the dark at 4°C until use.”

x) Line 110 However in this section the authors refer that Stock solution of TMPyP3 was prepared in demineralised (DEMI) water and diluted in (waste)water samples until the final concentration. All this seems a bit confuse. Why not always in the same conditions?

RE: The same stock solution of TMPyP3, prepared as described above, was used for all microbiological experiments. Different concentrations of TMPyP3 (as indicated in the manuscript under Materials and methods) were prepared in wastewater for the studies of spectroscopic properties and (photo)stability of TMPyP3. For the measurements of singlet oxygen production, porphyrins (TMPyP3, and TMPyP4 in new experiments) were initially dissolved in DMSO, to match the control dye – DPBF, which could only be dissolved in DMSO, and then diluted in ethanol. In all final solutions, whether singlet oxygen production was measured in DEMI water or wastewater (or also in tap water in new experiments), the percentage of DMSO present was less than 0.1%. All these subsections in the Materials and methods section are appropriately revised.

xi) In line 209 the authors refer that The photophysical and photochemical properties of TMPyP3 in the wastewater sample were evaluated by absorption and fluorescence spectroscopy, and (photo)stability, and singlet oxygen production (1O2) were measured to examine the influence of the inorganic and organic matter contained in water. However, in the experimental part the authors refer that before the experiments, wastewater was filtered and sterilised by autoclaving (121°C, 1.2 bar, 20 min). Please clarify this aspect and indicate the type of filter used.

RE: The wastewater sample was filtered to remove larger pieces of waste, and the Whatman No. 3 filter paper was used. Pore size of this filter paper is 6 �M, thus dissolved organic matter can easily pass through this filter. However, considering that autoclaving changes the physicochemical properties of wastewater, physicochemical analyses were carried out both before and after autoclaving, and these data are added to the Supporting information (S1 Table). We added information about the filter used and physicochemical properties of the wastewater sample before, and after filtration and autoclaving, so now it reads: “Before the experiments, wastewater was filtered using the Whatman No. 3 filter paper (Macherey-Nagel, Duren, Germany), and sterilised by autoclaving (121°C, 1.2 bar, 20 min). Physicochemical properties of sampled wastewater were determined before, and after filtration and sterilisation (S1 Table).” 

xii) In fig 4 is not clear in the Y axis that what is being observed is a decrease in fluorescence

RE: Name of the Y axis in Fig 4 is changed from % fluorescence to the decrease of fluorescence (%). In addition, a short explanation is added in the Materials and Methods section about the calculation of the singlet oxygen production, the results of which are shown in Fig 4.

xiii) In the discussion concerning the achievments already obtained with other porphyrins it would be more easy to follow if the authors introduce their structures in the same figure of TMPyP3.

RE: Structures of other porphyrins important for discussion (TMPyP3-CH3, TMPyP3-C17H35 and TMPyP4) are added to the same figure with TMPyP3.

xiv) During the discussion the authors must refer the previous figures presented in the results and related with the subject that they are commenting in order to facilitate their comprehension.

RE: We are thankful to the Reviewer for pointing this out. We have revised the discussion in accordance with this instruction.

xv) Line 402 The following comment is a bit confuse and must be clarified “should be noted that high photodegradation of DPBF without presence of TMPyP3 was also observed, especially in wastewater, which could be due to overlap of the applied irradiation wavelength (411 nm) with the absorption spectra of DPBF, and the resulting photochemical reaction that is enhanced in the presence of organic matter”

RE: We have revised this part of the text to be more precise, and this text now reads: “A significant photodegradation of DPBF without presence of TMPyP3 (in DPBF control) was observed, especially in wastewater (Fig 4). This could be due to the applied irradiation wavelength (411 nm), which is within the absorption spectrum of DPBF, so the dye itself can be a photosensitiser and produce ROS, while the organic matter present in wastewater contributes to this production even more.”

xvi) No errors can be visualized in graphics of figure 7

RE: Error bars are present in Fig 7, and there are very small standard deviations. We opted for this graphic because we wanted to compare the kinetics results for the performed measurements with different concentrations. All the obtained results and calculated average and SD values for Fig 7 (and also for Figures 5 and 6) are given in S4 Dataset, which is added in the Supporting information.

xviii) Line 466 The following comment is difficult to accept :Surprisingly, the high dark toxicity of TMPyP3 on K. pneumoniae was observed in tap water, but in the municipal wastewater sample, the sensitivity of both K. pneumoniae strains to TMPyP3 without light activations was much lower. It must be noted that during all our experiments it was not possible to exclude absolutely all light, and it was shown, e.g., with TMPyP4, that even ambient light during the experiment can be responsible for the observed ‘dark toxicity’, which was reported for several bacteria (47).” So the authors are trying to say that it is difficult to exclude all light so that toxicity is due to the photodynamic action of porphyrins? Usually the systems can be easily protected from light by covering with aluminium foil. Concerning the comment with TMPyP4, that even ambient light during the experiment can be responsible for the observed ‘dark toxicity’, which was reported for several bacteria what the authors want to say? If is at ambient light they can not say that it is observed dark toxicity.

RE: The reviewer is right about excluding light, and all our experiments, in which we tested dark toxicity, we have protected from light using aluminium foil. However, we have noticed that throughout the literature there is this important discussion about the fact that for photosensitizers that have a very high singlet oxygen production (such as with TMPyP4, and for TMPyP3 is even slightly higher, which was confirmed in our 1O2 production measurements), it is possible that even very low amount of light may lead to photoactivation. In all our dark toxicity experiments, the experiments were all the time maximally protected from light, but for example pipetting is difficult to carry out in 100% darkness, so we thought it was important to mention this in the discussion and cite the paper where this was emphasised. To clarify this aspect, the text in the manuscript is revised and now it reads: “It must be noted that during all our experiments, in which we tested toxicity in the dark, great care was taken to maximally protect them from light. However, it has been shown with strong 1O2 producers, e.g. with TMPyP4, that even very short exposure to ambient, even very low light, may be partially responsible for the observed "dark toxicity", which has been reported for several bacteria.” 

xix) The discussion is really hard to follow and to facilitate the reading the authors must introduce Figures 8 -10 in this section. It is not clear the real achievements obtained in these studies. So this section requires a real improvement.

RE: The discussion is now shortened and thoroughly revised. In the Figures 8 – 10 we wanted to compare the concentrations of TMPyP3 that have the most similar effect (log10 CFU/mL reduction values) on all three bacteria in the dark, and upon PDI, in tap water and wastewater. The aim was to further emphasize the advantage of photodynamic inactivation over water treatments without light. However, we decided to remove them because they burden the results and discussion sections, without a sufficiently new contribution to justify it.

xx) The conclusion must also be improved since does not reflect what was done and what was attained.

RE: The conclusion was revised to be clear and more precise, and to emphasize which studies were conducted, why, and what their main findings were.

Reviewer #2: General comments

In this study, Muskovic et al evaluated the effects of Antimicrobial Photodynamic Treatment (APDT) with a tetracationic porphyrin and violet-blue light on Klebsiella pneumonia and Pseudomonas aeruginosa bacteria, both in tap and wastewater. The results showed that APDT is able to inactivate planktonic cells of both species in both situations. The work is well written, presented and the results are discussed appropriately. However, the effects of APDT with various photosensitizers, including porphyrins, on these bacterial species, both in tap and wastewater, have already been described previously. Thus, the main weakness of the work is that the authors do not make it clear, from the beginning of the manuscript, what is the novelty of their studies and how they contribute to the advancement of APDT.

RE: We are thankful to the Reviewer for all the comments and suggestions according to which we have made changes to the manuscript as follows.

Abstract

Lines 15-17. Rewrite.

RE: This sentence is revised and now reads: “Photodynamic inactivation (PDI) that combines a photosensitiser and light in the presence of oxygen to generate singlet oxygen and other reactive species, which in turn react with a range of biomolecules, including the oxidation of bacterial genetic material, may be a way to stop the spread of antibiotic-resistant genes.”

Introduction

Lines 148-49. Rewrite.

RE: This sentence is revised and now reads: “Samples of DEMI and tap water for all experiments were sterilised using the same procedure as described above, and all water samples were kept at 4°C until use.”

Materials and Methods

Lines 130-133. Rewrite.

RE: This subsection is thoroughly revised.

Line 146. How was the filtration performed?

RE: The filtration was performed using the Whatman No. 3 filter paper. This information is added to the manuscript.

Lines 158, 165 and 180. How many biological replicates were performed in each of the experiments? This needs to be stated clearly.

RE: Information on the number of repetitions of microbiological experiments is added in each of these subsections. Moreover, S4 Dataset with all survival, dark toxicity and photoinactivation results for all three bacteria, with all calculated average and SD values is added in the Supporting information.

Results.

Line 299. Change to “continued to increase”

RE: Corrected.

Discussion

The discussion is too long and could be shortened. Lines 447-449. This is predominantly results (including Fig. 11) and should be moved to the “Results” section

RE: The discussion is shortened and revised. The sentence that describes Fig. 11 and the figure are moved to the ‘Results’.

Conclusions

Lines 478-479. The first sentence is not a conclusion and should be removed.

RE: The first sentence is removed and ‘Conclusions’ are revised.

Reviewer #3: Review of „Photoinactivation of multidrug-resistant strains of Klebsiella pneumoniae and Pseudomonas aeruginosa in municipal wastewater by tetracationic porphyrin and violet-blue light.” By Martina Musckovic et al. The paper describes the effect of cationic porphyrin (TMPyP3) activated with violet-blue light (VBL) on three strains of Gram-negative bacteria. The experiments were properly performed using standard techniques, and the conclusions are sound and correct. However, the novelty is not emphasized here. The authors performed photoinactivation experiments on liquid bacterial cultures dispersed in tap water and compared them to cells dissolved in wastewater, finding that all three strains could be photoinactivated with relatively low PS concentrations and low light doses. 

RE: We are thankful to the Reviewer for all the comments and suggestions. We emphasized the aim of our study and the main new findings, and made other modifications to the manuscript according to the reviewer’s comments as follows.

Although I believe that PDI could be used for wastewater treatment to prevent the spread of MDR bacteria, I feel it is a limitation of this manuscript to work with only one compound. 

RE: The aim of our study was on the one hand to investigate the influence of wastewater constituents on the properties of photosensitizer and PDI effect, and on the other hand to compare the sensitivity of different MDR bacterial strains to PDI. In this study, our goal was not to look for a better compound, so we did not compare TMPyP3 with another, but in the discussion, we looked at some of the known comparisons. Nevertheless, we added a comparison of singlet oxygen production between TMPyP3 and a much more researched regioisomer 5,10,15,20-(pyridinium-4-yl)porphyrin tetrachloride (TMPyP4). Since TMPyP4 is a very well-known and well-described photosensitiser, we used it as a reference for singlet oxygen production and for comparison with TMPyP3 in wastewater, but also in tap and DEMI water. In all these conditions, TMPyP3 showed somewhat better singlet oxygen production (added in Fig 4), which agrees with some other literature results that we cite in our paper. In addition, we determined the MEC values for TMPyP4 on all three bacterial strains in wastewater, under the same conditions as for TMPyP3 (with light irradiation for 10 min, and without light), and the obtained MEC values are also somewhat higher for TMPyP4 (S3 Table), in accordance with 1O2 production. We also tried to better clarify the aim of our study so we revised that part in the introduction. 

The experiments performed answer the question of whether TMPyP3-PDI kills bacteria but say nothing about the mechanism of bacterial death. Figure 11 would be a good starting point to explain the mechanism of bacterial death, but more is not known or discussed.

RE: Figure 11 (now Fig 8) is moved to the results section where is shortly described, and the discussion about the mechanism of bacterial death is added, with comments on this figure, in the last part of the discussion section.

I would expect the manuscript to be greatly shortened, especially the discussion section, which is largely a repetition of the results. Instead, the manuscript would benefit greatly if the authors compared the efficacy of the approach studied with others based on different PS. This could be done not necessarily based on experiments conducted by the authors but by other groups as well. Such a comparison would give the reader an idea of which properties of PS are beneficial in treating wastewater.

RE: The discussion is shortened and thoroughly revised, and it was taken into account that all the results presented in the previous section were discussed. Comparisons of our results with the results from the literature are also emphasized in the discussion.

Furthermore, it is not clear to me why the authors chose this particular PS for the studies.

RE: We explained in detail in the discussion why we chose TMPyP3 for this study, but we agree with the Reviewer that it should be even clearer and stated much earlier in the text, so we revised and moved this explanation to the last part of the introduction.

The abstract is too general, I would encourage the authors to be more specific, write a number instead of the term "high", state the concentration instead of writing "at the same concentration". It is not entirely clear what is meant here by "controlled fluences and light doses" (p. 2 lane 27-29). Is it not the case that scientists always use precise doses of light for a particular application?

RE: We are thankful to the Reviewer for these suggestions. Therefore, we added concrete concentrations to the abstract. We also revised the last sentence in the abstract to make it clearer what the aim of this study was, and now that part of the text reads: “This study confirmed the importance of studying the impact of water constituents on the properties and PDI effect of the applied photosensitiser, as well as checking the sensitivity of targeted bacteria to light of a certain wavelength, in conditions as close as possible to those in the intended application, to adjust all parameters and perfect the method.”

Authors use several expressions for photodynamic treatment: PDI, aPDT, PDT – that could be confusing.

RE: In the introduction, we listed the most frequently used terms and their abbreviations. We felt this was important because there is still a debate about which are the most appropriate and in which context.

It is not clear whether the authors use MDR to describe the phenomenon of multidrug resistance or strains that are multidrug resistant.

RE: We use it for multidrug-resistant strains.

‘ESKAPE bacteria and their biofilms’ – did you mean: bacteria and biofilm they produce?

RE: Yes, thank you for this correction. We have corrected accordingly in the manuscript. 

Authors also use terms: pre-treatment, primary and secondary treatment in relation to wastewater, but this is not explained at all and is unknown to non-specialist readers.

RE: A brief explanation is added and a new reference for a literature source where this can be found in great detail.

I would not fully agree with the statement that protein inactivation leads to ARG destruction (p. 3, pp. 67-69). Such a mental shortcut could be misleading. Amino acids and proteins are the first biomolecules exposed to ROS, probably because of their abundance in cells, and DNA and RNA could also be destroyed if PS comes close, however, destruction of ARGs is a side effect of DNA damage when it occurs but is not targeted.

RE: We completely agree with the Reviewer, thus we added a text (bold) to this sentence that generated ROS “react with biomolecules to which PS is close enough, and which may include lipids, amino acids and proteins, as well as nucleic acids…”

“Water applications" - unclear (p. 4 lane84).

RE: This sentence is revised and now reads: “Porphyrins are one of the most widely used groups of PSs for PDT and are also very suitable for various applications in water and water treatments.”

“Enterococcus" - did you mean Enterococcus spp. or the entire genus?

RE: In the paper we cited, Enterococci group was studied and the authors used the name Enterococcus as such.

Please provide an antibiotic resistance profile for the strains studied.

RE: The results of the conducted antibiotic sensitivity test for the strains studied are in S2 Table, which is added in Supporting information, and referred to in the Materials and methods section (under Bacterial strains and sample preparation).

Figures 8, 9 and 10 should be combined.

RE: The reviewer is right that these figures should be viewed together, given that, with comparisons in tap water and wastewater, our intention was to compare the concentrations of TMPyP3 that have the most similar effect on all three bacteria in the dark, and upon PDI. However, we decided to remove them because they burden the results and discussion sections, without a sufficiently new contribution to justify it.

Reviewer #4: In this study the authors report photodynamic effect of tetracationic water soluble porphyrin, 5,10,15,20-(pyridinium-3-yl) porphyrin tetrachloride (TMPyP3) using violet-blue light as light source, against of P. aeruginosa and K. pneumoniae in municipal wastewater. The aim of the work is highly ambitious, and the results are well explored and are, indeed, very promising. The use of aPDT in wastewater disinfection is the subject of several research groups, but has encountered some obstacles for its implementation in the field. These hindrances were mentioned by the authors, however, there are many points that need to be better discussed:

RE: We are thankful to the Reviewer for all the comments and suggestions according to which we have made changes to the manuscript as follows.

- It is not clear the main reason for the use of a violet-blue light source, mainly its advantage in practical field. It would not be more cost-effective the use of sunlight or even a white light source?

RE: We agree with the Reviewer that for the application of photoinactivation in wastewater disinfection, it would be best to use sunlight for photoactivation in terms of sustainability. However, in this study our aim was to evaluate the influence of wastewater constituents, and considering that the composition of wastewater is complex and can vary, we wanted to have all the other parameters specified and as simple as possible to monitor. Although the average fluence rate of solar irradiation can be obtained, we were interested in monitoring the kinetics of photoinactivation with different concentrations of TMPyP3 using the same fluence rate of a certain wavelength. We decided on a wavelength of 394 nm for several reasons, which we stated in the manuscript, and in the revised paper we tried to emphasize and clarify this even more. Violet-blue light is part of the solar spectrum, so its impact can be expected in applications with the use of solar irradiation, and it is the wavelength that penetrates the deepest through the water. Many exogenous PSs have good absorption at that wavelength, and we used it also for TMPyP3 in our previous work where we examined the influence of ions present in tap water on PDI, so here we could focus better on organic matter in wastewater. Finally, one of the aims in this study was to compare the susceptibility of different Gram-negative strains from the group of ESKAPE bacteria to PDI. One of the problems of existing water disinfectant methods is the different sensitivity of individual strains, which may lead to even greater resistance of those that are much less susceptible. Therefore, we chose Pseudomonas aeruginosa for comparison, which is known to be sensitive to violet-blue light alone (i.e. without the presence of exogenous PS), while on the other hand, we chose Klebsiella pneumoniae, which is known to be significantly less sensitive to violet-blue light.

- Authors referred organic and inorganic matter could be important factors in the photodynamic efficiency, however, no discuss are made to clarify this point. How can those chemicals interferer with photodynamic efficiency? The organic and inorganic municipal wastewater herein used was analyzed?

RE: One of the main aims of our study was to investigate the influence of wastewater constituents on the properties of photosensitizer and PDI effect. Given that in our previous two papers (which we refer to and cite in the discussion) we analysed the impact of inorganic ions present in tap water on TMPyP3 and its PDI effect, and since tap water is part of the wastewater, in this work we used tap water for comparison to emphasize the impact of organic matter present in wastewater. We added this clarification and revised that part accordingly in the introduction. In addition to the already conducted analysis of the physicochemical properties of wastewater, we have added the results of determining the physicochemical properties of wastewater after filtration and sterilization (S1 Table). In accordance with these results, we revised the discussion and tried to better emphasize the observed impact of wastewater constituents on PDI effect.

- It is not clear whether TMPy3 could be removed from the water after treatment. In a real application scenario, it is possible the use of a photostable and non-immobilized porphyrin as antimicrobial agent that will be released into natural waters? Do the authors have any data that could elucidate this point?

RE: In our study we did not examine this, but we cited research articles in the introduction and discussion where very similar compounds, most notably TMPyP4 (regioisomer of TMPyP3), were immobilized on different solid supports (cellulose, SiO2, glass, different polymers etc.), and TMPyP3 was previously used immobilised on cellulosic fabric. Another approach would be to remove TMPyP3 from water by adsorption on activated carbon filters, which we also mentioned in the introduction. The use of cheap activated carbon to remove cationic dyes was shown by Nizam et al. (Sci Rep 2021, 11, 8623, doi: 10.1038/s41598-021-88084-z). Given that TMPyP3 in our study proved to be quite stable in wastewater, it would certainly be better to examine this possibility first in future studies, and then monitor gradual photodegradation under solar radiation.

Reviewer #5: General comments

This manuscript describes some properties of a tetracationic porphyrin named TMPyP3 and the photoinactivation of Klebsiella pneumoniae and Pseudomonas aeruginosa in municipal wastewater irradiated with violet-blue light. The porphyrin used is a conventional photosensitizer studied for more than 25 years (eg see JPPB, 1996, 35, 149) in the PDI of microorganisms and other applications. In its present form, the manuscript is not uninteresting and does not add much to previous knowledge on the topic developed. This is also manifested in the ambiguous conclusions of the work.

RE: We are thankful to the Reviewer for all the comments and suggestions according to which we have made changes to the manuscript as follows. 

- Abstract. The first introductory sentences should be considerably shortened,

- Abstract. “tetracationic symmetric porphyrin” what does symmetric mean? It is unclear.

- Abstract. What does "(1,562 uM)” mean?

RE: The first sentences in the abstract are shortened. Instead of “tetracationic symmetric porphyrin” the full name of TMPyP3 is given (5,10,15,20-(pyridinium-3-yl)porphyrin tetrachloride). The concentration value is corrected with the appropriate unit (�M).

- Introduction. Several significant contributions in PDI with tetracationic porphyrins are missing.

- Introductio. “light, which also penetrates best through water.” Water does not absorb visible light. It should be explained.

RE: Several contributions in PDI with tetracationic porphyrins relevant to this work are added in introduction (including ref. JPPB, 1996, 35, 149). The sentence about the penetration of blue light into the water that was not clear enough has been revised and now reads: “Porphyrins are one of the most widely used groups of PSs for PDT and are also very suitable for various applications in water and water treatments. Their absorption is throughout the visible part of the electromagnetic (EM) spectrum, so they can be activated by artificial light as well as sunlight. Their particularly high absorption at the Soret band corresponds to blue light, which is the wavelength that penetrates the deepest in water.”

- The name of TMPyP3 should be correctly given the first time it is named and then use its abbreviation.Why was TMPyP3 selected as a photosensitizer? The studies should be compared with those obtained for active photosensitizers under the same experimental conditions.

RE: The full name of TMPyP3 is given (5,10,15,20-(pyridinium-3-yl)porphyrin tetrachloride) the first time it is named in the abstract. We explained in detail in the discussion why we chose TMPyP3 for this study, but we agree with the Reviewer that it should be even clearer and stated much earlier in the text, so we revised and moved this explanation to the introduction.

Furthermore, we added a comparison of singlet oxygen production between TMPyP3 and a much more researched regioisomer 5,10,15,20-(pyridinium-4-yl)porphyrin tetrachloride (TMPyP4). Since TMPyP4 is a very well-known and well-described photosensitiser, we used it as a reference for singlet oxygen production and for comparison with TMPyP3 in wastewater, but also in tap and DEMI water. In all these conditions, TMPyP3 showed somewhat better singlet oxygen production (added in Fig 4), which agrees with some other literature results that we cite in our paper. In addition, we determined the MEC values for TMPyP4 on all three bacterial strains in wastewater, under the same conditions as for TMPyP3 (with light irradiation for 10 min, and without light), and the obtained MEC values are also somewhat higher for TMPyP4 (S3 Table), in accordance with 1O2 production. Comparisons of our PDI results with the most relevant results from the literature are also emphasized in the discussion.

- How was exactly the stock solution of TMPyP3 prepared?

RE: This is revised in the Materials and methods section (under “Photosensitiser”), and now it reads. “A stock solution of TMPyP3 (200 µM) for microbiological experiments was prepared in sterile tap water (sterilised by autoclaving at 121°C, 1.2 bar, 20 min). After dissolving porphyrin in water, the solution was filtered using sterile Syringe Filter 0,45 �m (Labex Ltd, Budapest, Hungary), and kept covered in the dark at 4°C until use.”

- Absortion and fluorescence emission spectra should be compared with those previously reported in different media.

RE: We have discussed this, but in the revised discussion we have now moved it earlier in the text to emphasise better.

- In singlet oxygen determinations, the kinetic results should be given (60 s for 10 minutes.). Also, the spectral changes of DPFB should be shown in supporting information.

RE: The kinetics results in singlet oxygen determinations, for those obtained in DEMI and wastewater are added in the Supporting information section as S2 Dataset (“Kinetics of singlet oxygen production in DEMI water and municipal wastewater”), and for the results in tap water as S3 Dataset (“Singlet oxygen production in tap water”).

1,3-Diphenylisobenzofurane (DPBF), a fluorescent dye we have used for singlet oxygen determinations, reacts with 1O2 via [4+2]-cycloaddition forming a non-fluorescent endoperoxide product that yields dibenzoyl benzene (DBB), which also has different spectroscopic properties. So, the DPBF fluorescence decrease measured in our experiment is proportional to the formation of non-fluorescent product and to the amount of produced singlet oxygen (1O2). Fluorescence decrease was measured as a single value at λ = 453 nm after the excitation of the DPBF at λ = 420 nm. Excitation and emission detection peaks were determined measuring the whole spectra before measuring the 1O2 production. All 1O2 production measurements were performed in triplicate. No spectral changes, besides the decrease of fluorescence intensity due to reaction with 1O2 described above and measured as a single value per time point, did not occur. We have expanded the description of the experiment in the Materials and Methods section to further clarify these aspects.

- The production of singlet oxygen should be compared with another photosensitizer as a reference in the same medium.

RE: We have again conducted experiments to measure singlet oxygen production using 5,10,15,20-(pyridinium-4-yl)porphyrin tetrachloride (TMPyP4), a very well-known and well-described photosensitiser, as a reference. The results obtained for DPBF control, TMPyP3 and TMPy4 in DEMI water and wastewater are in Fig 4, and all the kinetics results are in the Supporting information (S2 Dataset). We also added the results for new experiment preformed with these compounds in tap water, and they are in the Supporting information (S3 Dataset).

- Figure 4 should be better described and explained in the text.

RE: The description and explanation of the Figure 4 is revised in the text.

- Table 1 and Table 2. What is the actual number of significant figures in the values?

RE: Minimal effective concentration (MEC) value, given in Table 1, was determined as the lowest concentration of porphyrin that reduces bacterial growth by 99.9%, and the minimal anti-adhesive concentration (MAAC) value, in Table 2, as the lowest concentration that affects the bacterial adhesion to the surface of polystyrene. The experiments to obtain the MEC and MAAC values were repeated, and exactly the same values as in the first experiment were obtained in the second, so there are no deviations.

- Figures 5-7. What is the statistical significance of the values? What is the error of each determination?

RE: S4 Dataset with all survival, dark toxicity and photoinactivation results for all three bacteria, shown in Figures 5, 6 and 7, with all calculated average and standard deviation values, is added in the Supporting information.

- If the purpose is to decontaminate wastewater from microorganisms, why was white light of natural origin not used?

RE: We are aware that for the application of photoinactivation in wastewater disinfection, it would be best to use sunlight for photoactivation in terms of sustainability. However, although numerous examples of antimicrobial PDT can be found in the literature, and this number is constantly growing, it is increasingly pointed out that the literature results of PDI are difficult to compare because different light sources, fluence rates and doses of light, together with different concentrations of photosensitisers (PSs) and different pathogens are used (Phasupan et al. 2021, Piksa et al. 2023). To compare the effect of PSs on different bacteria, it is useful to control the absorbed photons (Phasupan et al. 2021), and it is important to have as precise optical and biological parameters as possible in in vitro studies (Piksa et al. 2023) to better optimize PDI for different applications later and on different pathogens. White light as well as sunlight encompass a wide range of wavelengths of different intensity. In solar irradiation experiments, the average fluence rate can be obtained, but since we wanted to focus on the influence of wastewater constituents, and considering that the composition of wastewater is complex and can vary, in this study we wanted to have all the other parameters specified and as simple as possible to monitor. Therefore, we decided to monitor the kinetics of photoinactivation of different concentrations of TMPyP3 using the same fluence rate of a certain wavelength. We decided on a wavelength of 394 nm for several reasons, which we stated in the manuscript, and in the revised paper we tried to emphasize and clarify this even more. Violet-blue light is part of the solar spectrum, and it penetrates deeply through the water. Many exogenous PSs have good absorption at that wavelength, and also bacterial endogenous PSs, so we wanted to compare the susceptibility of different Gram-negative strains from the group of ESKAPE bacteria to PDI.

- How were the biofilms prepared?

RE: Biofilms were not studied, and therefore they were not prepared in our work. In our work minimal anti-adhesive concentration (MAAC) as the lowest concentration that affects the bacterial adhesion to the surface of polystyrene was determined, and an anti-adhesion effect was shown, suggesting the possibility of application in biofilm control.

- The discussion is extremely poor and lacks comparison with previous studies.

RE: The discussion is now revised and comparison with relevant previous studies is emphasised.

- Conclusions are unclear and imprecise.

RE: Conclusions are revised to be clear and more precise, and to emphasize which studies were conducted, why, and what their main findings were.

---

## [Decision Letter · Decision Letter 1]

1 Aug 2023

Photodynamic inactivation of multidrug-resistant strains of Klebsiella pneumoniae and Pseudomonas aeruginosa in municipal wastewater by tetracationic porphyrin and violet-blue light: the impact of wastewater constituents

PONE-D-23-13741R1

Dear Dr. Malatesti,

We’re pleased to inform you that your manuscript has been judged scientifically suitable for publication and will be formally accepted for publication once it meets all outstanding technical requirements.

Kind regards,

Adelaide Almeida

Academic Editor

PLOS ONE

Additional Editor Comments (optional):

Reviewers' comments:

Reviewer's Responses to Questions

**Comments to the Author**

1. If the authors have adequately addressed your comments raised in a previous round of review and you feel that this manuscript is now acceptable for publication, you may indicate that here to bypass the “Comments to the Author” section, enter your conflict of interest statement in the “Confidential to Editor” section, and submit your "Accept" recommendation.

Reviewer #1: All comments have been addressed

Reviewer #2: All comments have been addressed

Reviewer #4: All comments have been addressed

Reviewer #5: (No Response)

2. Is the manuscript technically sound, and do the data support the conclusions?

Reviewer #1: Yes

Reviewer #2: (No Response)

Reviewer #4: Yes

Reviewer #5: Yes

3. Has the statistical analysis been performed appropriately and rigorously? 

Reviewer #1: I Don't Know

Reviewer #2: (No Response)

Reviewer #4: Yes

Reviewer #5: Yes

4. Have the authors made all data underlying the findings in their manuscript fully available?

Reviewer #1: Yes

Reviewer #2: (No Response)

Reviewer #4: Yes

Reviewer #5: Yes

5. Is the manuscript presented in an intelligible fashion and written in standard English?

Reviewer #1: Yes

Reviewer #2: (No Response)

Reviewer #4: Yes

Reviewer #5: Yes

6. Review Comments to the Author

Reviewer #1: The authors address in general most of my comments.

Considering the NMR of the porphyrin the authors must indicate the multiplet by a range of values and not by a single value. In the title probably before by tetracationic porphyrin is better to introduce by a tetractionic porphyrin.

Reviewer #2: (No Response)

Reviewer #4: The authors clearly answered all the questions asked and completed or adjusted the manuscript to clarify some questions that raised some doubts. I recommend the publication in PlosOne.

Reviewer #5: Photoinactivation of multidrug-resistant strains of Klebsiella pneumoniae and

Pseudomonas aeruginosa in municipal wastewater by tetracationic porphyrin and

violet-blue light

Authors have adequately addressed my comments

7. PLOS authors have the option to publish the peer review history of their article (what does this mean?). If published, this will include your full peer review and any attached files.

Reviewer #1: No

Reviewer #2: **Yes: **Gilberto Úbida Leite Braga

Reviewer #4: No

Reviewer #5: No

---

## [Editor Report · Acceptance letter]

7 Aug 2023

PONE-D-23-13741R1 

Photodynamic inactivation of multidrug-resistant strains of *Klebsiella pneumoniae* and Pseudomonas aeruginosa in municipal wastewater by tetracationic porphyrin and violet-blue light: the impact of wastewater constituents 

Dear Dr. Malatesti:

I'm pleased to inform you that your manuscript has been deemed suitable for publication in PLOS ONE. Congratulations! Your manuscript is now with our production department. 

Kind regards, 

on behalf of

Dr. Adelaide Almeida 

Academic Editor

PLOS ONE